# Space-Time Continuous PDE Forecasting using Equivariant Neural Fields

**David M. Knigge**[*,1], **David R. Wessels**[*,1], **Riccardo Valperga**[1], **Samuele Papa**[1], **Jan-Jakob Sonke**[2],
**Efstratios Gavves** [†,1], **Erik J. Bekkers** [†,1]
[1]University of Amsterdam     [2]Netherlands Cancer Institute
d.m.knigge@uva.nl, d.r.wessels@uva.nl

## Abstract

Recently, Conditional Neural Fields (NeFs) have emerged as a powerful modelling paradigm for PDEs, by learning solutions as flows in the latent space of the Conditional NeF. Although benefiting from favourable properties of NeFs such as grid-agnosticity and space-time-continuous dynamics modelling, this approach limits the ability to impose known constraints of the PDE on the solutions – e.g. symmetries or boundary conditions – in favour of modelling flexibility. Instead, we propose a space-time continuous NeF-based solving framework that - by preserving geometric information in the latent space - respects known symmetries of the PDE. We show that modelling solutions as flows of pointclouds over the group of interest $G$ improves generalization and data-efficiency. We validated that our framework readily generalizes to unseen spatial and temporal locations, as well as geometric transformations of the initial conditions - where other NeF-based PDE forecasting methods fail - and improve over baselines in a number of challenging geometries.

## 1   Introduction

Partial Differential Equations (PDEs) are a foundational tool in modelling and understanding spatio-temporal dynamics across diverse scientific domains. Classically, PDEs are solved using numerical methods such as finite elements, finite volumes, or spectral methods. In recent years, Deep Learning (DL) methods have emerged as promising alternatives due to abundance of observed and simulated data as well as the accessibility to computational resources, with applications ranging from fluid simulations and weather modelling [51, 7] to biology [33].

---

* shared first author, † shared lead advising

38th Conference on Neural Information Processing Systems (NeurIPS 2024).

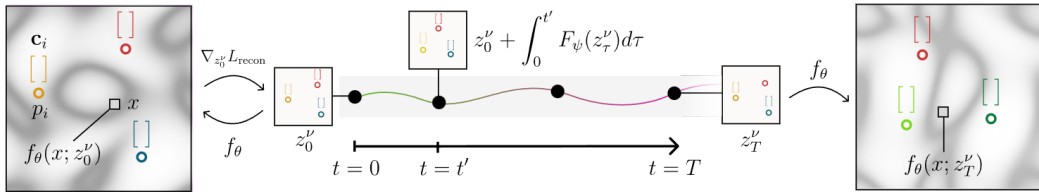

Figure 1: We propose to solve an equivariant PDE in function space by solving an equivariant ODE in latent space. Through our proposed framework, which leverages *Equivariant Neural Fields* $f_\theta$, a field $\nu_t$ is represented by a set of latents $z_t^\nu = \{(p_i^\nu, \mathbf{c}_i^\nu)\}_{i=1}^N$ consisting of a *pose* $p_i$ and context vector $\mathbf{c}_i$. Using meta-learning, the initial latent $z_0^\nu$ is fit in only 3 SGD steps, after which an equivariant neural ODE $F_\psi$ models the solution as a latent flow.

The systems modelled by PDEs often have underlying symmetries. For example, heat diffusion or fluid dynamics can be modeled with differential operators which are rotation equivariant, e.g., given a solution to the system of PDEs, its rotation is also a valid solution [1]. In such scenarios it is sensible, and even desirable, to design neural networks that incorporate and preserve such symmetries to improve generalization and data-efficiency [12, 48, 4].

Crucially, DL-based approaches often rely on data sampled on a regular grid, without the inherent ability to generalize outside of it, which is restrictive in many scenarios [40]. To this end, [51] propose to use Neural Fields (NeFs) for modelling and forecasting PDE dynamics. This is done by fitting a neural ODE [11] to the conditioning variables of a conditional Neural Field trained to reconstruct states of the PDE [13]. However, this approach fails to leverage aforementioned known symmetries of the system. Furthermore, using neural fields as representations has proved difficult due to the non-linear nature of neural networks [13, 3, 35], limiting performance in more challenging settings. We posit that NeF-based modelling of PDE dynamics benefits from representations that account for the symmetries of the system as this allows for introducing inductive biases into the model that ought to be reflected in solutions. Furthermore, we show that through meta-learning [28, 45] the NeF backbone improves performance for complex PDEs by further structuring the NeF's latent space, simplifying the task of the neural ODE.

We introduce a framework for *space-time continuous equivariant PDE solving*, by adapting a class of $\mathrm{SE}(n)$-Equivariant Neural Fields (ENFs) to PDE-specific symmetries. We leverage the ENF as representation for modelling spatiotemporal dynamics. We solve PDEs by learning a flow in the latent space of the ENF - starting at a point $z_0$ corresponding to the initial state of the PDE - with an equivariant graph-based neural ODE [11] we develop from previous work [5]. We extend the ENF to equivariances beyond $\mathrm{SE}(n)$, by extending its weight-sharing scheme to equivalance classes for specific symmetries relevant to our setting. Furthermore, we show how meta-learning [14, 28, 45, 13], can not only significantly reduce inference time of the proposed framework, but also substantially simplify the structure of the latent space of the ENF, thereby simplifying the learning process of the latent dynamics for the neural ODE model. We present the following contributions:

- We introduce a framework for spatio-temporally continuous PDE solving that respects known symmetries of the PDE through equivariance constraints.

- We show that correctly chosen equivariance constraints as inductive bias improves performance of the solver - in terms of MSE - in spatio-temporally continuous settings, i.e. evaluated *off* the training grid and beyond the training horizon.

- We show how meta-learning improves the structure of the latent space of the ENF, simplifying the learning process, leading to better performance in solving PDEs.

We structure the paper as follows: in Sec. 2 we provide an overview of the mathematical preliminaries and describe the problem setting. Our proposed framework is introduced in Sec. 3. We validate our framework on different PDEs defined over a variety of geometries in Sec. 4, with differing equivariance constraints, showing competitive performance over other neural PDE solvers.We provide an in-depth positioning of our approach in relation to other work in Appx. A.

## 2   Mathematical background and problem setting

**Continuous spatiotemporal dynamics forecasting.**   The setting considered is data-driven learning of the dynamics of a system described by continuous observables. In particular, we consider flows of fields, denoted with $\hat{\nu} : \mathbb{R}^d \times [0, T] \to \mathbb{R}^c$. We use $\hat{\nu}_t$ as a shorthand for $\hat{\nu}(\cdot, t)$. We assume the flow is governed by a PDE, and consider the Initial Value Problem (IVP) of predicting $\hat{\nu}_t$ from a given $\nu_0$. The dataset consists of field snapshots $\nu : \mathcal{X} \times [\![T]\!] \to \mathbb{R}^c$, in which $[\![T]\!] := 1, 2, \ldots, T$ denotes the set of time points on which the flow is sampled and $\mathcal{X} \subset \mathbb{R}^d$ is a set of coordinate values. For each time point we are given a set of input-output pairs $[\mathcal{X}, \nu(\mathcal{X})]$ where $\nu(\mathcal{X}) \subset \mathbb{R}^c$ are the values of the field at those coordinates. Importantly, the location at which the field is sampled need not be regular, i.e., we do not require the training data to be on a grid or to be regularly spaced in time, nor need coordinate values be identical for train and test sets. Following [51], we distinguish between $t_{\text{in}}$ - referring to values within the training time horizon $[0, T]$ - and $t_{\text{out}}$ - analogously to values beyond $T$.

---

[1]Assuming boundary conditions are symmetric, i.e. they transform according to the relevant group action.

**Neural Fields in dynamics modelling.** Conditional Neural fields (NeFs) are a class of coordinate-based neural networks, often trained to reconstruct discretely-sampled input continuously. More specifically, a conditional neural field $f_\theta : \mathbb{R}^n \to \mathbb{R}^d$ is a field –parameterized by a neural network with parameters $\theta$– that maps input coordinates $x \in \mathbb{R}^n$ in the data domain alongside conditioning latents $z$ to $d$-dimensional signal values $\nu(x) \in \mathbb{R}^d$. By associating a conditioning latent $z^\nu \in \mathbb{R}^c$ to each signal $\nu$, a single conditional NeF $f_\theta : \mathbb{R}^n \times \mathbb{R}^c \to \mathbb{R}^d$ can learn to represent families $\mathcal{D}$ of continuous signals such that $\forall \nu \in \mathcal{D} : f(x) \approx f_\theta(x; z^\nu)$. [51] propose to use conditional NeFs for PDE modelling by learning a continuous flow in the latent space of a conditional neural field. In particular, a set of latents $\{z_i^\nu\}_{i=1}^T$ are obtained by fitting a conditional neural field to a given set of observations $\{\nu_i\}_{i=1}^T$ at timesteps $1, ..., T$; simultaneously, a neural ODE [11] $F_\psi$ is trained to map pairs of temporally contiguous latents s.t. solutions correspond to the trajectories traced by the learned latents. Though this approach yields impressive results for sparse and irregular data in planar PDEs, we show it breaks down on complex geometries. We hypothesize that this is due to lack of a latent space that preserves relevant geometric transformations that characterize the symmetries of the systems we are modelling, and as such propose an extension of this framework where such symmetries are preserved.

**Symmetries and weight sharing.** Given a group $G$ with identity element $e \in G$, and a set $X$, a *group action* is a map $\mathcal{T} : G \times X \to X$. For simplicity, we denote the action of $g \in G$ on $x \in X$ as $gx := \mathcal{T}(g, x)$, and call *G-space* a smooth manifold equipped with a $G$ action. A group action is homomorphic to $G$ with its group product, namely it is such that $ex = x$ and $(gh)x = g(hx)$. As an example, we are interested in the Special Euclidean group $\mathrm{SE(n)} = \mathbb{R}^n \rtimes SO(n)$: group elements of SE(n) are identified by a translation $t \in \mathbb{R}^n$ and rotations $\mathbf{R} \in SO(n)$ with group operation $gg' = (t, \mathbf{R}_\theta)(t', \mathbf{R}_{\theta'}) = (\mathbf{R}\mathbf{x}' + \mathbf{x}, \mathbf{R}\mathbf{R}_{\theta'})$; We denote by $\mathcal{L}_g$ the left action of $G$ on function spaces defined as $\mathcal{L}_g f(\mathbf{x}') = f(g^{-1}\mathbf{x}') = f(\mathbf{R}_\theta^{-1}(\mathbf{x}' - \mathbf{x}))$. Many PDEs are defined by *equivariant* differential operators such that for a given state $\nu$: $\mathcal{L}_g \mathcal{N}[\nu] = \mathcal{N}[\mathcal{L}_g \nu]$. If the boundary conditions do not break the symmetry, namely if the boundary is symmetric with respect to the same group action, then a $G$-transformed solution to the IVP for some $\nu_0$ corresponds to the solution for the $G$-transformed initial value. For example, laws of physics do not depend on the choice of coordinate system, this implies that many PDEs are defined by SE(n)-equivariant differential operators. The *geometric deep learning* literature shows that models can benefit from leveraging the inherent symmetries or invariances present in the data by constraining the searchable function space through *weight sharing* [9, 25, 5]. Recall that in our framework we model flows of fields, solutions to PDEs defined by equivariant differential operators, with ordinary differential equations in the latent space of conditional neural fields. We leverage the symmetries of the system for two key aspects of the proposed method: first by making the relation between signals and corresponding latents equivariant; second, by using equivariant ODEs, namely ODEs defined by equivariant vector fields: if $\frac{dz}{d\tau} = F(z)$ is such that $F(gz) = gF(z)$, then solutions are mapped to solutions by the group action.

## 3 Method

We adapt the work of [51], and consider the following optimization problem [2]:

$$\min_{\theta, \psi, z_\tau} \quad \mathbb{E}_{\nu \in D, x \in \mathcal{X}, t \in [\![T]\!]} \|\nu_t(x) - f_\theta(x; z_t^\nu)\|_2^2, \quad \text{where} \quad z_t^\nu = z_0^\nu + \int_0^t F_\psi(z_\tau^\nu)d\tau, \quad (1)$$

with $f_\theta(x; z_t^\nu)$ a decoder tasked with reconstructing state $\nu_t$ from latent $z_t^\nu$ and $F_\psi$ a neural ODE that maps a latent to its temporal derivative: $\frac{dz_\tau^\nu}{d\tau} = F_\psi(z_\tau^\nu)$, modelling the solution as flow in latent space starting at the initial latent $z_0^\nu$ - see Fig. 1 for a visual intuition.

**Equivariant space-time continuous dynamics forecasting.** A PDE defined by a $G$-equivariant differential operator - for which $\mathcal{L}_g \mathcal{N}[\nu] = \mathcal{N}[\mathcal{L}_g \nu]$ - are such that solutions are mapped to other solutions by the group action if the boundary conditions are symmetric. We would like to leverage this property, and *constrain* the neural ODE $F_\psi$ such that the solutions it finds in latent space can be mapped onto each other by the group action. Our motivation for this is twofold: (1) it is natural for

---

[2]We highlight that [51] optimize latents $z_t^\nu$, neural field $f_\theta$, and ODE $F_\psi$ using two separate objectives. We instead found that our framework is more stable under single-objective optimization.

our model to have, by construction, the geometric properties that the modelled system is known to posses - (2) to get more structured latent representations and facilitate the job of the neural ODE. To achieve this we first need the latent space $Z$ to be equipped with a well-defined group action with respect to which $\forall g \in G, z \in Z : F_\psi(gz) = gF_\psi(z)$, and, most importantly, we need the relation between the reconstructed field and the corresponding latent to be equivariant, i.e.,

$$\forall g \in G\,,\ x \in \mathcal{X} : \mathcal{L}_g f_\theta(x; z_t^\nu) = f_\theta(g^{-1}x; z_t^\nu) = f_\theta(x; gz_t^\nu). \tag{2}$$

Note that, somewhat imprecisely, we call this condition *equivariance* to convey the idea even though it is not, strictly speaking, the commonly used definition of equivariance for general operators. If we consider the decoder as a mapping from latents to fields, we can make the notion of equivariance of this mapping more precise. Namely

$$f(x) = D_\theta(z), D_\theta(z) : z_t^\nu \mapsto f_\theta(\cdot; z_t^\nu), f(g^{-1}x) = D_\theta(gz), D_\theta(gz) : g\, z_t^\nu \mapsto f_\theta(g^{-1}\cdot; z_t^\nu)\,. \tag{3}$$

In Sec. 3.1 we describe the Equivariant Neural Field (ENF)-based decoder, which satisfies equation (2). Second, in Sec. 3.2 we outline the graph-based equivariant neural ODE. Sec. 3.3 explains the motivation for- and use of- meta-learning for obtaining the ENF backbone parameters. We show how the combination of equivariance and meta-learning produce much more structured latent representations of continuous signals (Fig. 3).

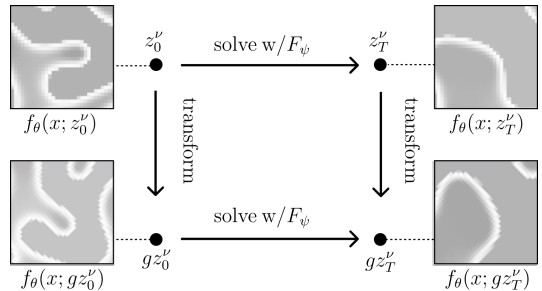

### 3.1 Representing
### PDE states with Equivariant Neural Fields

We briefly recap ENFs here, referring the reader to [49] for more detail. We extend ENFs to symmetries for PDEs over varying geometries.

Figure 2: The proposed framework respects pre-defined symmetries of the PDE: a rotated solution $\mathcal{L}_g \nu_T$ may be obtained either by solving from latent $z_0^\nu$ (top-left) and transforming the solution $z_T^\nu$ (top-right) to $gz_T^\nu$ (bottom-right) or transforming $z_0^\nu$ to $gz_0^\nu$ (bottom-left) and solving this.

**ENFs as cross-attention over bi-invariant attributes.** Attention-based conditional neural fields represent a signal $\nu \in \mathcal{D}$ with a corresponding *latent set* $z^\nu$ [52]. This class of conditional neural fields obtain signal-specific reconstructions $\nu(x) \approx f_\theta(x; z^\nu)$ through a cross-attention operation between the latent set $z^\nu$ and input coordinates $x$. ENFs [49] extend this approach by imposing equivariance constraints w.r.t a group $G \subseteq SE(n)$ on the relation between the neural field and the latents such that transformations to the signal $\nu$ correspond to transformation of the latent $z^\nu$ (Eq. (2)). For this condition to hold, we need a well-defined action on the latent space $Z$ of $f_\theta$. To this end, ENFs define elements of the latent set $z^\nu$ as tuples of *pose* $p_i \in G$ and *context* $\mathbf{c}_i \in \mathbb{R}^d$, $z^\nu := \{(p_i, \mathbf{c}_i)\}_{i=1}^N$. The latent space is then equipped with a group action defined as $gz = \{(gp_i, \mathbf{c}_i)\}_{i=1}^N$. To achieve equivariance over transformations ENFs follow [5] where equivariance is achieved with convolutional *weight-sharing* over equivalence classes of points pairs $x, x'$. ENFs instead extend weight-sharing to cross-attention over *bi-invariant* attributes of $z, x$ pairs.

Weight-sharing over bi-invariant attributes of $z, x$ is motivated by Eq. 2, by which we have:

$$f_\theta(x; z) = f_\theta(gx; gz). \tag{4}$$

Intuitively, the above equation says that a transformation $g$ on the domain of $f_\theta$, i.e. $g^{-1}x$, can be undone by *also* acting with $g$ on $z$. In other words, the output of the neural field $f_\theta$ should be *bi-invariant* to $g-$transformations of the pair $z, x$. For a specific pair $(z_i, x_m) \in Z \times X$, the term bi-invariant attribute $\mathbf{a}_{i,m}$ describes a function $\mathbf{a} : (z_i, x_m) \mapsto \mathbf{a}(z_i, x_m)$ such that $\mathbf{a}(z_i, x_m) = \mathbf{a}(gz_i, gx_m)$. Throughout the paper we use $\mathbf{a}_{i,m}$ as shorthand for $\mathbf{a}(z_i, x_m)$.

To parameterize $f_\theta$, we can accordingly choose any function that is bi-invariant to $G-$transformations of $z, x$. In particular, for an input coordinate $x_m$ ENFs choose to make $f_\theta$ a cross-attention operation between attributes $\mathbf{a}_{i,m}$ and the invariant context vectors $\mathbf{c}_i$:

$$f_\theta(x_m, z) = \text{cross\_attn}(\mathbf{a}_{:,m}, \mathbf{c}_:, \mathbf{c}_:) \tag{5}$$

As an example, for SE(n)-equivariance, we can define the bi-invariant simply using the group action: $\mathbf{a}_{i,m}^{SE(n)} = p_i^{-1} x_m = \mathbf{R}_i^T(x_m - x_i)$, which is bi-invariant by:

$$\forall g \in SE(n):\ (p_i, x) \mapsto (g\, p_i, g\, x)\ \Leftrightarrow\ p_i^{-1}x \mapsto (g\, p_i)^{-1} g\, x = p_i^{-1} g^{-1} g\, x = p_i^{-1} x\,. \tag{6}$$

**Bi-invariant attributes for PDE solving.** As explained above, ENF is equivariant to $SE(n)$-transformations by defining $f_\theta$ as a function of an $SE(n)-$bi-invariant attribute $\mathbf{a}^{SE(n)}$. Although many physical processes adhere to roto-translational symmetries, we are also interested in solving PDEs that - due to the geometry of the domain, their specific formulation, and/or their boundary conditions - are not fully $SE(n)-$equivariant. As such, we are interested in extending ENFs to equivariances that are not strictly (subsets of) $SE(n)$, which we show we can achieve by finding bi-invariants that respect these particular transformations. Below, we provide two examples, the other invariants we use in the experiments - including a "bi-invariant" $\mathbf{a}^\emptyset$ that is not actually bi-invariant to any geometric transformations, which we use to ablate over equivariance constraints - are in Appx. D.

*The flat 2-torus.* When the physical domain of interest is continuous and extends indefinitely, periodic boundary conditions are often used, i.e. the PDE is defined over a space topologically equivalent to that of the 2-torus. Such boundary conditions break $SO(2)$ symmetries; assuming the domain has periodicity $\pi$ and none of the terms of this PDE depend on the choice of coordinate frame, these boundary conditions imply that the PDE is equivariant to periodic translations: the group of translations modulo $\pi$: $\mathbb{T}^2 \equiv \mathbb{R}^2/\mathbb{Z}^2$. In this case, periodic functions over $x, y$ with periods $\pi$ would work as a bi-invariant, i.e. using poses $p \in \mathbb{T}^2$, $\mathbf{a}^{\mathbb{T}^2} = \cos(2\pi(x_0 - p_0)) + \cos(2\pi(x_1 - p_1))$ - which happens to be bi-invariant to rotations by $\frac{\pi}{2}$ as well. Instead, since we do not assume any rotational symmetries to exist on the torus, we opt for a non-rotationally symmetric function:

$$\mathbf{a}_{i,m}^{\mathbb{T}^2} = \cos(2\pi(x_i^0 - p_i^0)) \oplus \cos(2\pi(x_i^1 - p_i^1)), \tag{7}$$

where $\oplus$ denotes concatenation. This bi-invariant is used in experiments on Navier-Stokes over the flat 2-Torus.

*The 2-sphere.* In some settings a PDE may be symmetric only to rotations along a certain axes. An example is that of the global shallow-water equations on the two-sphere - used to model geophysical processes such as atmospheric flow [16], which are characterised by rotational symmetry only along the earth's axis of rotation due to inclusion of a term for Coriolis acceleration that breaks full $SO(3)$ equivariance. We use poses $p \in SO(3)$ parametrised by Euler angles $\phi, \theta, \gamma$, and spherical coordinates $\phi, \theta$ for $x \in S^2$. We make the first two Euler angles coincide with the spherical coordinates and define a bi-invariant for rotations around the axis $\theta = \pi$.

$$\mathbf{a}_{i,m}^{SW} = \Delta\phi_{p_i, x_m} \oplus \theta_{p_i} \oplus \gamma_{p_i} \oplus \theta_{x_m}, \tag{8}$$

where $\Delta\phi_{p_i, x_m} = \phi_{p_i} - \phi_{x_m} - 2\pi$ if $\phi_{p_i} - \phi_{x_m} > \pi$ and $\Delta\phi_{p_i, x_m} = \phi_{p_i} - \phi_{x_m} + 2\pi$ if $\phi_{p_i} - \phi_{x_m} < -\pi$, to adjust for periodicity.

In summary, to parameterize an ENF equivariant with respect to a specific group we are simply required to find attributes that are bi-invariant with respect to the same group. In general we achieve this by using group-valued poses and their action on the PDE domain.

## 3.2 PDE solution as latent space flow

Let $z_0^\nu$ be a latent set that faithfully reconstructs the initial state $\nu_0$. We want to define a neural ODE $F_\psi$ that map latents $z_t^\nu$ to their temporal derivatives $\frac{dz_\tau^\nu}{d\tau} = F_\psi(z_\tau^\nu)$ that is equivariant with respect to the group action: $gF_\psi(z_\tau^\nu) = F_\psi(gz_\tau^\nu)$. To this end, we use a *message passing neural network* (MPNN) to learn a flow of poses $p_i$ and contexts $\mathbf{c}_i$ over time. We base our architecture on P$\Theta$NITA [5], which employs convolutional weight-sharing over bi-invariants for $SE(n)$. For an in-depth recap of message-passing frameworks, we refer the reader to Appx. A. Since $F_\psi$ is required to be equivariant w.r.t. the group action, any updates to the poses $p_i$ should also be equivariant. [41] propose to parameterize an equivariant node position update by using a basis spanned by relative node positions $x_j - x_i$. In our setting, poses $p_i$ are points on a manifold $M$ equipped with a group action. As such, we analogously propose parameterizing pose updates by a weighted combination of logarithmic maps $\log_{p_i}(p_j)$, which intuitively describe the relative position between $p_i, p_j$ in the tangent space $T_{p_i}M$, or the displacement from $p_i$ to $p_j$. We integrate the resulting pose update over the manifold through the exponential map $\exp_{p_i}$. In the euclidean case $\log_{p_i}(p_j) = x_j - x_i$ and we get back node position updates per [41]. In short, the message passing layers we use consist of the

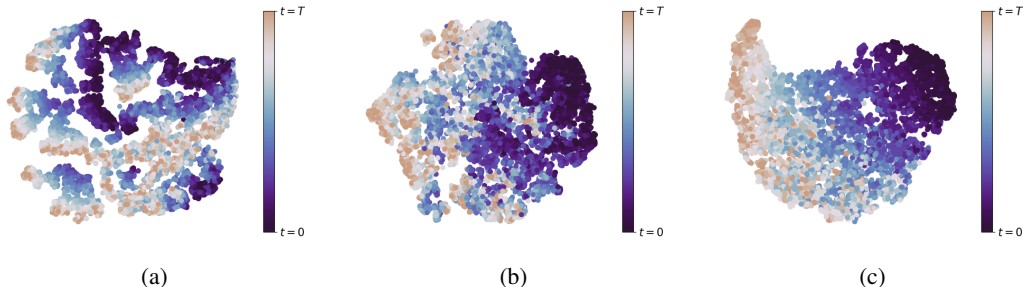

Figure 3: We show the impact of meta-learning and equivariance on the latent space of the ENF when representing trajectories of PDE states. Fig. 3a shows a T-SNE plot of the latent space of $f_\theta$ when $z_t^\nu$ is optimized with autodecoding, and no weight sharing over bi-invariants is enforced. Fig. 3b shows the latent space when meta-learning is used, but no weight sharing is enforced. Fig. 3c shows the latent space when $z_t^\nu$ are obtained using meta-learning and $f_\theta$ shares weights over $\mathbf{a}^{\mathrm{SE(n)}}$.

following update functions:

$$\mathbf{c}_i^{l+1} = \sum_{(p_j, \mathbf{c}_j) \in z^{\nu, l}} k^{\text{context}}(\mathbf{a}_{i,j}^l)\mathbf{c}_j^l, \quad p_i^{l+1} = \exp_{p_i^l}\left(\frac{1}{N}\sum_{(p_j^l, \mathbf{c}_j^l) \in z^{\nu, l}} k^{\text{pose}}(\mathbf{a}_{i,j}^l)\mathbf{c}_j^l \log_{p_i^l}(p_j^l)\right),$$

(9)

with $k^{\text{context}}, k^{\text{pose}}$ message functions weighting the incoming context and pose updates, parameterized by a two-layer MLP as a function of the respective bi-invariant.

### 3.3 Obtaining the initial latent $z_0^\nu$

Until now we've not discussed how to obtain latents corresponding to the initial condition $z_0^\nu$. An approach often used in conditional neural field literature is that of autodecoding [36], where latents $z^\nu$ are optimized for reconstruction of the input signal $\nu$ with SGD. Optimizing a NeF for reconstruction does not necessarily lead to good quality representations [35], i.e. using MSE-based autodecoding to obtain latents $z_t^\nu$ - as is proposed by [51] - may complicate the latent space, impeding optimization of the neural ODE $F_\psi$. Moreover, autodecoding requires many optimization steps at inference (for reference, [51] use 300-500 steps). [13] propose meta-learning as a way to overcome long inference times, as it allows for fitting latents in a few steps - typically three or four. We hypothesize that meta-learning may also structure the latent space - similar to the impact of equivariance constraints, since the very limited number of optimization steps requires efficient organization of latents $z_t^\nu$ around the (shared) initialization, forcing together the latent representation of contiguous states. To this end, we propose to use meta-learning for obtaining the initial latent $z_0^\nu$, which is then unrolled by the neural ode $F_\psi$ to find solutions $z_t^\nu$.

### 3.4 Equivariance and meta-learning structure the latent space $Z$

As a first validation of the hypotheses that both equivariance constraints and meta-learning introduce structure to the latent space of $f_\theta$, we visualize latent spaces of different variants of the ENF. We fit ENFs to a dataset consisting of solutions to the heat equation for various initial conditions (details in Appx. E). For each sample $\nu_t$, we obtain a set of latents $z_t^\nu$, which we average over the invariant context vectors $\mathbf{c}_i \in \mathbb{R}^c$ to obtain a single vector in $\mathbb{R}^c$ invariant to a group action according to the chosen bi-invariant. Next, we apply T-SNE [47] to the resulting vectors in $\mathbb{R}^c$. We use three setups: (a) no meta-learning, $\theta$ and latents $z_t^\nu$ optimized for every $\nu_t$ separately using autodecoding [36], and no equivariance imposed (per Eq. 15), shown in Fig. 3a. (b) meta-learning is used to obtain $\theta, z_t^\nu$, but no equivariance imposed, shown in Fig. 3b and (c) meta-learning is used to obtain $\theta, z_t^\nu$ and SE(2)-equivariance is imposed by weight-sharing over $\mathbf{a}^{\mathrm{SE(n)}}$ bi-invariants, shown in Fig. 3c. The results confirm our intuition that both meta-learning and equivariance improve latent-space structure.

**Recap: optimization objective.** We use a meta-learning inner-loop [28, 13] to obtain the initial latent $z_0^\nu$ under supervision of coordinate-value pairs $(x, \nu(x)_0)_{x \in \mathcal{X}}$ from $\nu_0$. This latent is unrolled for $t_{\text{train}}$ timesteps using $F_\psi$. The obtained latents are used to reconstruct states $z_t^\nu$ along the trajectory

of $\nu$, and parameters of $f_\theta, F_\psi$ are optimised for reconstruction MSE, as shown in the left-hand side of Eq. 1. See Appx. B for detailed pseudocode of this process.

## 4 Experiments

We intend to show the impact of symmetry-preservation in continuous PDE solving. To this end we perform a range of experiments assessing different qualities of our model on tasks with different symmetries. First, we investigate the **equivariance properties** of our framework by evaluating it against unseen geometric transformations of the initial conditions. Next, we assess **generalization** and **extrapolation** capabilities w.r.t. unseen spatial locations and time horizons inside and outside the time ranges seen during training respectively, **robustness** to partial test-time observations, and **data-efficiency**. As the continuous nature of NeF-based PDE solving allows, we verify these properties for PDEs defined over **challenging geometries**: the plane $\mathbb{R}^2$, 2-torus $\mathbb{T}^2$ and the sphere $S^2$ and the 3D ball $\mathbb{B}^3$. Architectural details and hyperparameters are in Appx. E. *Code is available on GitHub.*

Additionally, we validate our model on a benchmark of PDEs that exhibits **no transformation symmetries**: the CFDBench [32] benchmark. We include details on parameter counts, memory usage and runtimes of our model compared to baselines in Appx. F.

### 4.1 Datasets and evaluation

All datasets are obtained by randomly sampling disjoint sets of initial conditions for train and test sets, and solving them using numerical methods. Dataset-specific details on generation can be found in Appx. E. •**Heat equation on $\mathbb{R}^2$ and $S^2$.** The heat equation describes diffusion over a surface: $\frac{dc}{dt} = D\nabla^2 c$, where $c$ is a scalar field, and $D$ is the diffusivity coefficient. We solve it on the 2D plane where $\nabla^2 c = \frac{\partial^2 c}{\partial x_1} + \frac{\partial^2 c}{\partial x_2}$ - and on the 2-sphere $S^2$ where in spherical coordinates: $\nabla^2 c = \left( \frac{1}{\sin\theta} \frac{\partial}{\partial\theta} \left( \sin\theta \frac{\partial c}{\partial\theta} \right) + \frac{1}{\sin^2\theta} \frac{\partial^2 c}{\partial\phi^2} \right)$. Although a relatively simple PDE, we find that defining it over a non-trivial geometry such as the sphere proves hard for non-equivariant methods. •**Navier-Stokes on $\mathbb{T}^2$.** We solve 2D Navier Stokes [43] for an incompressible fluid with dynamics $\frac{dv}{dt} = -u\nabla v + v\Delta\mu + f, v = \nabla \times u, \nabla u = 0$, where $u$ is the velocity field, $v$

Table 1: MSE $\downarrow$ for heat equation on $\mathbb{R}^2$.

| | $t_{\text{IN}}$ TRAIN | $t_{\text{OUT}}$ TRAIN | $t_{\text{IN}}$ TEST | $t_{\text{OUT}}$ TEST |
|---|---|---|---|---|
| DINo [51] | 5.92E-04 | 2.40E-04 | 3.85E-03 | 5.12E-03 |
| Ours $a^\theta$ | **6.23±1.01E-06** | **4.90±20.1E-06** | 2.19±0.32E-03 | 5.08±13.2E-04 |
| Ours $a^{\text{SE}(2)}$ | 1.18±0.45E-05 | 2.53±3.50E-05 | **1.50±0.77E-05** | **2.53±3.43E-05** |

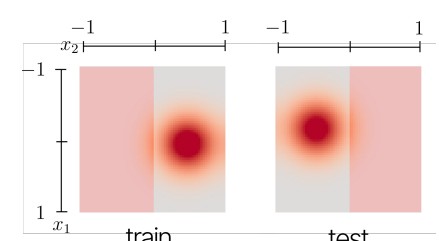

Figure 4: A train and test sample from the planar diffusion dataset. Initial conditions for train and test are spikes in disjoint subsets of $\mathbb{R}^2$.

the vorticity, $\mu$ the viscosity and $f$ a forcing term (see Appx. E). We create a dataset of solutions for the vorticity using Gaussian random fields as initial conditions. Due to the incompressibility condition, it is natural to solve this PDE with periodic boundary conditions corresponding to the topology of a 2-Torus $\mathbb{T}^2$ - implying equivariance to periodic translation. •**Shallow-water on $\mathbb{S}^2$.** The global shallow-water equations model large-scale oceanic and atmospheric flow on the globe, derived from Navier-Stokes under assumption of shallow fluid depth. The global shallow-water equations (see Appx. E) include terms for Coriolis accelleration, which makes this problem equivariant to rotation along the globe's axis of rotation. We follow the IVP specified by [16], and create a dataset of paired vorticity-fluid height solutions. •**Internally-heated convection in a 3D ball.** We solve the Boussinesq equation for internally heated convection in a ball, a model relevant for example in the context of the Earth's mantle convection. It involves continuity equations for mass conservation, momentum equations for fluid flow under pressure, viscous forces and buoyancy, and a term modelling heat transfer. We generate initial conditions varying the internal temperature using $N(0,1)$ noise and obtain solutions for the temperature defined over a regular spherical $\phi, \theta, r$ grid. •**CFDBench** [32] consists of a set of one-step solutions (pairs of input and output states $\nu_t, \nu_{t+1}$) for classic computational fluid dynamics (CFD) problems, with varying fluid properties, boundary conditions and geometries. The goal of this dataset is to assess generalizability of DL-based PDE solvers over problem parameters. This dataset does not exhibit transformation symmetries because of the absolute position of obstacles in the geometry and orientation of the flow.

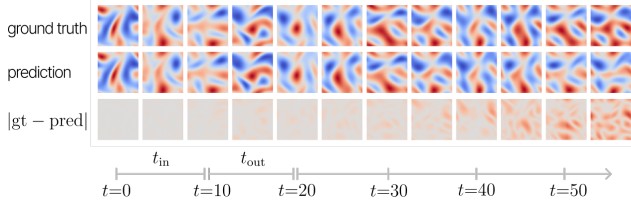

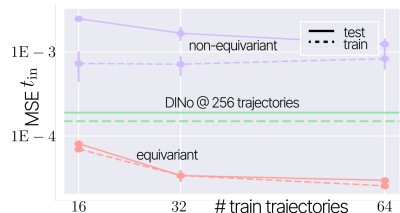

Figure 5: A Navier-Stokes test sample (top) and corresponding predictions from our model (bottom). We visualize predictions in the train horizon $t_{\text{in}} = [0, ..., 9]$, $t_{\text{out}} = [10, ..., 20]$ and beyond. The model remains stable well beyond the train horizon, but due to accumulated errors fails to capture dynamics beyond $t > 40$.

Figure 6: Test MSE $t_{\text{in}}$ for increasing training set sizes for the heat equation over the sphere. Equivariant improves over non-equivariant. For reference we show performance of DINo [51] trained on 256 trajectories.

**Evaluation.** All reported MSE values are for predictions obtained given only the initial condition $v_0$, with std over 3 runs. We evaluate two settings for train and test sets both: **generalization setting** with time evolution happening within the seen horizon during training ($t_{\text{in}}$); and, **extrapolation setting** with the time evolution happening outside the seen horizon during training ($t_{\text{out}}$). For both cases we measure the mean-squared error (MSE). To position our work relative to competitive data-driven PDE solvers, on the 2D-Navier-Stokes experiment we provide comparisons with a range of baselines. In most other settings these models cannot straightforwardly be applied, and we only compare to [51], to our knowledge the only other fully continuous PDE solving method in literature. For the Navier-Stokes and Internally-Heated Convection experiments, we compare with Transolver [50], which has shown SOTA results as DL-based PDE solving method for general geometries.

**Equivariance properties - heat equation on the plane.** To verify our framework respects the posed equivariance constraints, we create a dataset of solutions to the heat equation that *requires* a neural solver to respect equivariance constraints to achieve good performance. Specifically, for initial conditions we randomly insert a pulse of variable intensity in $x = (x_1, x_2) \in \mathbb{R}^2$ s.t. $-1 < x_1 < 1, 0 < x_2 < 1$ for the training data and $-1 < x_1 < 1, -1 < x_2 < 0$ for the test data. Intuitively,

train and test sets contain spikes under different disjoint sets of roto-translations (see Fig. 4). We train variants of our framework with ($\mathbf{a}^{\text{SE}(2)}$, Eq. 6) and without ($\mathbf{a}^{\emptyset}$, Eq. 15) equivariance constraints. In this dataset, we set $t_{\text{in}} = [0, ..., 9]$, and evaluation horizon $t_{\text{out}} = [10, ..., 20]$. Results in Tab. 1 show that the non-equivariant model, as well as the baseline [51] are unable to successfully solve test initial conditions, whereas the equivariant model performs well.

**Robustness to subsampling & time-horizons - Navier-Stokes on the 2-Torus.** We perform an experiment assessing the impact of equivariance constraints and meta-learning on robustness to sparse test-time observations of the initial condition. To this end, we train a model with ($\mathbf{a}^{\mathbb{T}^2}$, Eq. 7), without ($\mathbf{a}^{\emptyset}$, Eq. 15) equivariance constraints, and one with equivariance constraints and without meta-learning (AD $\mathbf{a}^{\mathbb{T}^2}$, Eq. 7), on a fully-observed train set. The training horizon $t_{\text{in}} = [0, ..., 9]$, and evaluation horizon $t_{\text{out}} = [10, ..., 20]$. Subsequently, we apply the trained model to the problem of solving from sparse initial conditions $v_0$, with observation

Table 2: MSE $\downarrow$ for Navier-Stokes on $\mathbb{T}^2$.

| | $t_{\text{IN}}$ TRAIN | $t_{\text{OUT}}$ TRAIN | $t_{\text{IN}}$ TEST | $t_{\text{OUT}}$ TEST |
|---|---|---|---|---|
| | 100% OF $\nu_0$ OBSERVED | | | |
| CNODE [2] | 6.02E-02 | 3.35E-01 | 5.48E-02 | 3.17E-01 |
| FNO | 9.43E-05 | 2.11E-03 | 8.44E-05 | 1.60E-03 |
| G-FNO | **3.13E-05** | **3.49E-04** | **3.15E-05** | **3.52E-04** |
| Transolver [50] | 1.80E-02 | 4.85E-01 | 1.85E-02 | 4.90E-01 |
| DINo [51] | 8.20E-03 | 6.85E-02 | 1.11E-02 | 9.08E-02 |
| Ours AD,$\mathbf{a}^{\mathbb{T}_2/\pi}$ | 5.60$_{\pm0.43}$E-02 | 0.37$_{\pm0.34}$E-01 | 6.75$_{\pm0.62}$E-02 | 4.00$_{\pm0.38}$E-01 |
| Ours $\mathbf{a}^{\emptyset}$ | 1.41$_{\pm1.83}$E-02 | 1.67$_{\pm1.27}$E-02 | 2.60$_{\pm3.16}$E-02 | 2.14$_{\pm1.46}$E-01 |
| Ours $\mathbf{a}^{\mathbb{T}_2/\pi}$ | 1.45$_{\pm0.08}$E-03 | 9.14$_{\pm0.36}$E-02 | 1.57$_{\pm0.09}$E-03 | 1.16$_{\pm0.14}$E-02 |
| | 50% OF $\nu_0$ OBSERVED | | | |
| CNODE [2] | 1.38E-01 | 6.33E-01 | 1.52E-01 | 6.76E-01 |
| FNO | 3.31E-02 | 1.39E-01 | 3.20E-02 | 1.47E-01 |
| G-FNO | 2.75E-02 | 1.17E-01 | 2.32E-02 | 1.01E-01 |
| Transolver [50] | 4.69E-01 | 0.99E-01 | 4.76E-01 | 0.99E-01 |
| DINo [51] | 3.67E-02 | 2.81E-01 | 3.74E-02 | 2.83E-01 |
| Ours AD,$\mathbf{a}^{\mathbb{T}_2/\pi}$ | 6.89$_{\pm2.68}$E-02 | 3.95$_{\pm2.18}$E-01 | 7.01$_{\pm3.56}$E-02 | 4.01$_{\pm2.29}$E-01 |
| Ours $\mathbf{a}^{\emptyset}$ | 1.05$_{\pm0.04}$E-02 | 1.45$_{\pm0.01}$E-01 | 2.60$_{\pm3.16}$E-02 | 2.14$_{\pm1.46}$E-01 |
| Ours $\mathbf{a}^{\mathbb{T}_2/\pi}$ | **1.50$_{\pm0.17}$E-03** | **8.97$_{\pm1.57}$E-03** | **5.75$_{\pm2.58}$E-03** | **5.03$_{\pm2.63}$E-02** |
| | 5% OF $\nu_0$ OBSERVED | | | |
| CNODE [2] | 1.23E+00 | 2.14E+00 | 1.20E+00 | 4.35E+00 |
| FNO | 4.13E-01 | 7.70E-01 | 3.84E-01 | 7.07E-01 |
| G-FNO | 3.56E-01 | 7.09E-01 | 3.40E-01 | 6.47E-01 |
| Transolver [50] | 8.48E-01 | 1.23E+00 | 8.28E-01 | 1.24E+00 |
| DINo [51] | 3.67E-02 | 2.81E-01 | 3.94E-02 | 2.91E-01 |
| Ours AD,$\mathbf{a}^{\mathbb{T}_2/\pi}$ | 6.89$_{\pm2.68}$E-02 | 3.95$_{\pm2.18}$E-01 | 7.01$_{\pm3.56}$E-02 | 4.01$_{\pm2.29}$E-01 |
| Ours $\mathbf{a}^{\emptyset}$ | 7.31$_{\pm1.37}$E-02 | 2.97$_{\pm2.42}$E-01 | 7.96$_{\pm1.65}$E-02 | 3.35$_{\pm3.41}$E-01 |
| Ours $\mathbf{a}^{\mathbb{T}_2/\pi}$ | **3.19$_{\pm1.07}$E-02** | **1.33$_{\pm0.35}$E-01** | **3.44$_{\pm1.43}$E-02** | **1.61$_{\pm4.93}$E-01** |

Table 3: Zero-shot temporal superresolution on Navier-Stokes.

| | $t_{\text{IN}}$ TEST | | | $t_{\text{OUT}}$ TEST | | |
|---|---|---|---|---|---|---|
| | $d\tau=1.0$ (train) | $d\tau=0.5$ | $d\tau=0.25$ | $d\tau=1.0$ (train) | $d\tau=0.5$ | $d\tau=0.25$ |
| DINo [51] | 1.19E-02 | 3.85E-02 | 3.98E-02 | 8.82E-02 | 2.19E-01 | 2.22E-01 |
| Ours | **1.76E-03** | **1.86E-03** | **1.91E-03** | **1.42E-02** | **1.49E-02** | **1.52E-02** |

rates where 50% and 5% of the initial condition is observed (Tab. 2). Approaches operating on discrete (CNODE [2]) and regular grids (FNO [29], G-FNO [20]) perform very well when evaluated on fully-observed regular grids, outperforming continuous approaches (ours, [51]). However, we note

that all discrete/regular models greatly deteriorate in performance when observation rates decrease. Equivariance constraints and meta-learning clearly improve performance overall, achieving best perfomance in all sparse settings. Our proposed framework performs competitively to discrete baselines and other NeF based PDE solving methods [51] in the fully observed setting. To qualitatively assess long-term stability well-beyond the train horizon, we visualize test trajectory and the solution found by our model for $t_{\text{in}} = [0, ..., 9]$, $t_{\text{out}} = [10, ..., 20]$ and beyond in Fig. 5. We show error accumulation for solutions on $100\%$ observed and $50\%$ observed initial states in Appx. F. To validate the time-continuous nature of our model, we train a model with supervision at a resolution of $d\tau{=}1.0$, and subsequently evaluate on resolutions $d\tau{=}0.5$ and $d\tau{=}0.25$ (Tab. 3), showing that our framework does not accumulate significant error when deploying on higher temporal resolution.

**Data-efficiency - Diffusion on the sphere.** To assess the impact of equivariance on data efficiency, we vary the size of the training set of heat equation solutions from 16 to 64 trajectories and apply a model with ($\mathbf{a}^{\text{SO}(3)}$, Eq. 13) and without ($\mathbf{a}^{\emptyset}$, Eq. 15) equivariance constraints. In this dataset, we set $t_{\text{in}} = [0, ..., 9]$, and evaluation horizon $t_{\text{out}} = [10, ..., 20]$. We visualize $t_{\text{in}}$ test- and train MSE in Fig. 6. These results show the non-equivariant model overfitting the training set for smaller numbers of trajectories while unable to solve the PDE satisfactorily, whereas the equivariant model generalizes well even with only 16 training trajectories.

**Super-resolution - Shallow-Water on the sphere.** Due to their continuous nature, NeF-based approaches inherently support zero-shot super-resolution. In this setting, we generate a set of solutions for the global shallow-water equations over $\mathbb{S}^2$ at $2\times$ resolution, and apply mean-pooling with a kernel size of 2 to obtain a low-resolution dataset. We train a model that respects rotational symmetries along the rotation axis of the globe ($\mathbf{a}^{\text{SW}}$, Eq. 8) at train resolution, and evaluate the model by solving initial conditions at $2\times$ resolution (Tab. 4, Fig. 7). In this dataset, we set $t_{\text{in}} = [0, ..., 9]$, and evaluation horizon $t_{\text{out}} = [10, ..., 14]$. First, we note that our model has difficulty capturing the dynamics near $t_{\text{out}}$ - and beyond the training horizon, i.e. $t => 9$ - we suspect because of accumulation of reconstruction errors impacting the ability of $F_{\psi}$ to model the relatively volatile dynamics of these equations. This points to a drawback of NeF-based solvers: error accumulation *starts* with the reconstruction error on the initial condition. Ranging over our experiments, we found that this error can be reduced by increasing model capacity, at steep cost of computational complexity attributable to the global attention operator in the ENF backbone. Regarding super-resolution; the model is able to solve the high-resolution initial conditions without inducing significantly increased MSE - it does not produce significant artefacts in the process.

**Challenging geometries - Internally heated convection in 3D ball.** We show the value of inductive biases in modelling over a challenging

Table 4: MSE $\downarrow$ on Shallow-Water equations on the sphere.

| | $t_{\text{IN}}$ TRAIN | $t_{\text{OUT}}$ TRAIN | $t_{\text{IN}}$ TEST | $t_{\text{OUT}}$ TEST |
|---|---|---|---|---|
| | TRAIN RESOLUTION | | | |
| DINo [51] | 1.75E-04 | **1.36E-03** | 2.01E-04 | **1.37E-03** |
| Ours $\mathbf{a}^{\text{SW}}$ | **9.94$_{\pm0.41}$E-05** | 1.89$_{\pm0.03}$E-03 | **1.09$_{\pm1.14}$E-04** | 1.87$_{\pm0.04}$E-03 |
| | ZERO-SHOT 2X SUPER-RESOLUTION | | | |
| DINo [51] | 3.03E-04 | 2.03E-03 | 3.37E-04 | 2.03E-03 |
| Ours $\mathbf{a}^{\text{SW}}$ | **1.58$_{\pm0.02}$E-04** | **1.96$_{\pm0.02}$E-03** | **1.61$_{\pm0.01}$E-04** | **1.93$_{\pm0.02}$E-03** |

ground truth low-res

ground truth high-res

zero-shot prediction

$t{=}0 \qquad t{=}3 \qquad t{=}6 \qquad t=11$

Figure 7: Test samples at train resolution (top), $2\times$ train resolution (middle) and corresponding predictions from our equivariant model ($\mathbf{a}^{\text{SW}}$ Eq. 8 (bottom). The model does not produce significant upsampling artefacts, but fails to capture dynamics outside the training horizon.

Table 5: MSE $\downarrow$ on Internally-Heated Convection in the ball.

| | $t_{\text{IN}}$ TRAIN | $t_{\text{OUT}}$ TRAIN | $t_{\text{IN}}$ TEST | $t_{\text{OUT}}$ TEST |
|---|---|---|---|---|
| | 100% OF $\nu_0$ OBSERVED | | | |
| Transolver [50] | **3.89E-04** | 1.88E-02 | **4.13E-04** | 2.09E-02 |
| DINo [51] | 2.94E-03 | 7.56E-02 | 3.06E-03 | 7.78E-02 |
| Ours $\mathbf{a}^{\mathbb{B}^3}$ | 5.79$_{\pm0.17}$E-04 | **7.72$_{\pm0.55}$E-03** | 5.99$_{\pm0.15}$E-04 | **7.97$_{\pm0.46}$E-03** |
| | 50% OF $\nu_0$ OBSERVED | | | |
| Transolver [50] | 4.39E-01 | 4.99E-01 | 4.38E-01 | 5.00E-01 |
| DINo [51] | 3.01E-03 | 1.06E-01 | 3.02E-03 | 1.13E-01 |
| Ours $\mathbf{a}^{\mathbb{B}^3}$ | **6.27E-04** | **7.76E-03** | **6.63E-04** | **8.21E-03** |

geometry. We apply an equivariant model ($\mathbf{a}^{\mathbb{B}^3}$, Eq. 14) to a set of solutions to Boussinesq internally heated convection in a ball defined over a regular $\phi, \theta, r$-grid, where we set $t_{\text{in}} = [0, ..., 9]$, and evaluation horizon $t_{\text{out}} = [10, ..., 14]$. Results (Tab. 6, Fig. 8) for our equivariant model show good generalization compared to a non-equivariant baseline [51]. The significantly larger Transolver [50] model (see Appx. F) obtains better performance within the train time horizon on train and test

sets, but overfits to this time horizon, generalizing poorly beyond. We additionally show results for sparsely observed input states, showing the limitations of non NeF-based solvers to generalize over irregular changing observation grids. We interpret these results as an indication of a marked reduction in solving-complexity and improved generalization when correctly accounting for a PDE's symmetries.

**Non-symmetric PDEs** Lastly, we evaluate our model in the setting when solving PDEs that do not exhibit any global symmetries to assess whether the preservation of symmetries in the latent space of our model precludes application to non-symmetric PDEs. We train a translation-equivariant ($\mathbf{a}^{\mathbb{R}^2}$) model, i.e. one with equivariance properties identical to a CNN, on the Cavity, Dam and Cylinder flows from CFDBench [32]. Comparing to the baselines set by the dataset authors, we see our method improves significantly over a number of classical baselines, despite no global transformational symmetries being present in the problems being solved.

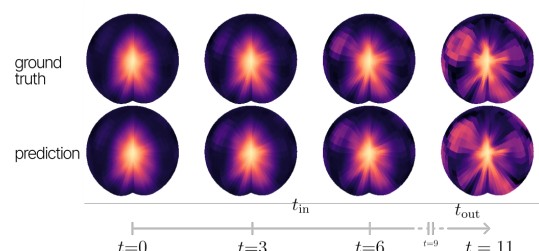

Figure 8: Test samples (top) and corresponding predictions from our model equivariant to $S^2$-rotations in the ball. (Eq. 14)

Table 6: Test MSE ↓ for CFDBench auto-regressive one forward propagation considering all properties for the cavity, dam and cylinder flows. Baselines taken from [32].

|  | CAVITY | DAM | CYLINDER |
|---|---|---|---|
| Identity | 6.42E-02 | 1.69E-03 | 7.54E-02 |
| Auto-FFN | 6.42E-02 | 1.68E-03 | 7.54E-02 |
| Auto-DeepONet | 6.39E-02 | 1.64E-03 | 7.53E-02 |
| Auto-EDeepONet | 6.45E-02 | 1.49E-03 | 7.43E-02 |
| Auto-DeepONetCNN | 6.33E-02 | 1.68E-03 | 7.52E-02 |
| FNO | 2.61E-02 | 8.75E-05 | 1.15E-03 |
| U-Net | 1.58E-02 | 1.70E-03 | 5.49E-05 |
| Ours $\mathbf{a}^{\mathbb{R}_2}$ | **1.10E-02** | **8.19E-05** | **1.42E-05** |

## 5 Limitations & Future work

We're interested in exploring the application of our ENF-based PDE solving framework to larger-scale, more complex problems. Throughout our experiments with ENFs we noticed that more complex signals, e.g. higher resolution PDEs, may be fit easily either by increasing the ENF hidden size or by increasing the number of latents used. However, either of these changes significantly impacts computational complexity, due to the calculation of attention coefficients in the latent space of the ENF for every latent-input coordinate pair. A possible way of addressing this is detailed in [49]; we can approximate the output of the attention operation through limiting the number of latents attended to (using k-nearest neighbours). This may open the door to modelling more complex, larger-scale dynamics than learned in present experiments.

## 6 Conclusion

We introduce a novel equivariant space-time continuous framework for solving partial differential equations (PDEs). Uniquely - our method handles sparse or irregularly sampled observations of the initial state while respecting symmetry-constraints and boundary conditions of the underlying PDE. We clearly show the benefit of symmetry-preservation over a range of challenging tasks, where existing methods fail to capture the underlying dynamics.

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

# A  Related work

**DL approaches to dynamics modelling**   In recent years, the learning of spatiotemporal dynamics has been receiving significant attention, either for modelling interacting systems [31, 30], scientific Machine Learning [51, 8, 7, 38, 26, 53], or even videos [1]. Most DL methods for solving PDEs attempt to directly replace solvers with mappings between finite-dimensional Euclidean spaces, i.e. through the use of CNNs [19, 2] or GNNs [39, 8] often applied autoregressively to an observed (discretized) PDE state. Instead, the Neural Operator (NO) [27] paradigm attempts to learn infinite-dimensional operators, i.e. mappings between function spaces, with limited success. Fourier Neural Operator (FNO) [29] extends this method by performing convolutions in the spectral domain. FNO obtains much improved performance, but due to its reliance on FFT is limited to data on regular grids.

**Inductive biases in DL and dynamics modelling**   Geometric Deep Learning aims to improve model generalization and performance by constraining/designing a model's space of learnable functions based on geometric principles. Prominent examples include Group Equivariant Convolutional Networks and Steerable CNNs [12, 4], generalizations of CNNs that respect symmetries of the data - such as dilations and continuous rotations [48, 15, 25]. Analogously, Graph Neural Networks (GNNs) [24] or Message Passing Neural Networks (MPNNS) [17] are a variant of neural network that respects set-permutations naturally found in graph data. They are typically formulated for graphs $\mathcal{G} = (\mathcal{V}, \mathcal{E})$, with nodes $i \in \mathcal{V}$ and edges $\mathcal{E}$. Typically nodes are embedded into a node vector $f_i^0$, which is subsequently updated over multiple layers of *message passing*. Message passing consists of (1) computing messages $m_{i,j}$ over edges $i, j$ from node $j$ to $i$ with the message function (taking into account edge attributes $a_{i,j}$: $m_{i,j} = \phi_m(f_i^l, f_j^l, a_{i,j})$ (2) aggregating incoming messages: $m_i = \sum_{j \in \mathcal{N}(i)} m_{i,j}$, (3) computing updated node features $f_i^{l+1} = \phi_u(f_i^l, m_i)$.

Recently, such methods have also been adapted for sparse physical data, e.g. for molecular property prediction [41, 6] - where the GNN is additionally required to respect transformation symmetries. [5] unifies these approaches to equivariance under the guise of *weight sharing* over equivalence classes defined by bi-invariant attributes of pairs of nodes $i, j$, a viewpoint we leverage in constructing the equivariant conditioning latent $z_t^\nu$ corresponding to a PDE state $\nu_t$. In the context of dynamics modelling, equivariant architectures have been employed to incorporate various properties of physical systems in the modelling process, examples of such properties are the symplectic structure [22], discrete symmetries such as reversing symmetries [46] and energy conservation [18, 21].

**Neural Fields in dynamics modelling**   Conditional Neural fields (NeFs) are a class of coordinate-based neural networks, often trained to reconstruct discretely-sampled input continuously. More specifically, a conditional neural field $f_\theta : \mathbb{R}^n \to \mathbb{R}^d$ is a field –parameterized by a neural network with parameters $\theta$– that maps input coordinates $x \in \mathbb{R}^n$ in the data domain alongside conditioning latents $z$ to $d$-dimensional signal values $f(x) \in \mathbb{R}^d$. By associating a conditioning latent $z^\nu \in \mathbb{R}^c$ to each signal $\nu$, a single conditional NeF $f_\theta : \mathbb{R}^n \times \mathbb{R}^c \to \mathbb{R}^d$ can learn to represent families $\mathcal{D}$ of continuous signals such that $\forall \nu \in \mathcal{D} : f(\mathbf{x}) \approx f_\theta(\mathbf{x}; z^\nu)$. [13] showed the viability of using the latents $\mathbf{z}^i$ as representations for downstream tasks (e.g. classification, generation) proposing a framework for *learning on neural fields*. This framework inherits desirable properties of neural fields, such as inherent support for sparsely and/or irregularly sampled data, and independence to signal resolution. [51] propose to use conditional NeFs for PDE modelling by learning a continuous flow in the latent space of a conditional neural field. In particular, a set of latents $\{\mathbf{z}_i^\nu\}_{i=1}^T$ are obtained by fitting a conditional neural field to a given set of observations $\{\nu_i\}_{i=1}^T$ at timesteps $1, ..., T$; simultaneously, a neural ODE [11] $F_\psi$ is trained to map pairs of temporally continuous latents s.t. solutions correspond to the trajectories traced by the learned latents. Though this approach yields impressive results for sparse and irregular data in planar PDEs, we show it breaks down on more challenging geometries. We hypothesize that this is due to a lack of a latent space that preserves relevant geometric transformation with respect to which systems we are modelling are symmetric, and as such propose an extension of this framework where such symmetries are preserved.

**Obtaining Neural Fields representations**   Most NeF-based approach to representation or reconstruction use SGD to optimize (a subset of) the parameters of the NeF, inevitably leading to significant overhead in inference; conditional NeFs require optimizing a (set of) latents from initialization to reconstruct for a novel sample. Accordingly, research has explored ways of addressing this limitation. [42, 45] propose using Meta-Learning [14, 34] to optimize for an initialization for the NeF from which it is possible to reconstruct for a novel sample in as few as 3 gradient descent steps. [13] propose to meta-learn the NeF backbone, but fix the initialization for the latent $\mathbf{z}$ and instead optimize the learning rate used in its optimization using Meta-SGD [28]. Recently, work has also explored the relation between initialization/optimization of a NeF and its value as downstream representation; [35] show that (1) using a shared NeF initialization and (2) limiting the number of gradient updates to the NeF improves performance in downstream tasks, as this simplifies the complex relation between a NeFs parameter space and its output function space. We combine these insights and make Meta-Learning part of our equivariant PDE solving pipeline, as it enables fast inference and we show it to simplify the latent space of the ENF, improving performance of the neural ODE solver.

# B  Pseudocode for optimization objective

See Alg. 1 for pseudocode of the training loop that we use, written for a single datasample for simplicity of notation. For simplicity, we further assume we're using an euler stepper to solve the neural ODE, but this can be replaced by any solver. For inference, this stratagem is identical, except we do not perform gradient updates to $\theta, \psi$.

---
**Algorithm 1** Optimization objective
---
Randomly initialize neural field $f_\theta$
Randomly initialize neural ode $F_\psi$
**while** not done **do**
 Sample initial states and coordinates $\nu_0$.
 Initialize latents $z_0^\nu \leftarrow \{(p_i, \mathbf{c}_i)\}_{i=1}^N$.
 **for all** step $\in N_{\text{initial state opt}} = 3$ **do**
  $z_0^\nu \leftarrow z_0^\nu - \epsilon \nabla_{z_0^\nu} \mathcal{L}_{\text{mse}}\big(f_\theta(\cdot, z_0^\nu), \nu_0)\big)$
 **end for**
 **for all** $t \in [1, ..., t_{\text{in}}]$ **do**
  $z_t^\nu \leftarrow z_0^\nu + \int_0^t F_\psi(z_\tau^\nu)d\tau$
 **end for**
Update $\theta, \psi$ per:
$\theta \leftarrow \theta - \eta \nabla_\theta \mathcal{L}'_{\text{mse}}, \psi \leftarrow \psi - \eta \nabla_\psi \mathcal{L}'_{\text{mse}}$ with $\mathcal{L}'_{\text{mse}} = \big(\big\{f_\theta(\cdot, z_t^\nu), \nu_t\big\}_{t=0}^{t_{\text{in}}}\big)$
**end while**
---

# C  Equivariant Neural Fields

**ENF to reconstruct PDE states**  For ease of notation we denote $\mathbf{P}$ and $\mathbf{C}$ the matrices containing poses and corresponding appearances stacked row-wise, i.e. $\mathbf{P}_{i,:} = p_i^T$ and $\mathbf{C}_{i,:} = \mathbf{c}_i^T$. Furthermore, we denote $\mathbf{A}$ as the matrix containing all bi-invariants $\mathbf{a}_{i,m}$ stacked row-wise, i.e. $\mathbf{A}_{i,:} = \mathbf{a}_{i,m}^T$:

$$f_\theta(\mathbf{x}; z^{\nu_t}) := \text{softmax}\left(\frac{\mathbf{Q}(\mathbf{A})\mathbf{K}^T(\mathbf{C})}{\sqrt{d_k}} + \mathbf{G}(\mathbf{A})\right)\mathbf{V}(\mathbf{C}; \mathbf{A}), \tag{10}$$

where the softmax is applied over the latent set and with $d_k$ the hidden dimensionality of the ENF. The query matrix $\mathbf{Q}$ is constructed as $\mathbf{Q}=\mathbf{W}_q \gamma_q^T(\mathbf{A})$, $\gamma_q$ a Gaussian RFF embedding [44], followed by a linear layer $\mathbf{W}_q$, i.e. $\mathbf{Q}$ consists of the RFF embedded bi-invariants of the input coordinate $x_m$ and each of the latent poses $p_i$ stacked row-wise. The key matrix is given by a learnable linear transformation $\mathbf{W}_k$ of the context vectors $\mathbf{c}_i$: $\mathbf{K}=\mathbf{W}_k\mathbf{C}^T$. The attention coefficients which result from the inner product of $\mathbf{Q}, \mathbf{K}$ are weighted by a Gaussian window $\mathbf{G}$ whose magnitude is conditioned on a distance measure on the relative distance between latent poses and input coordinates as: $\mathbf{G}_i = \sigma_{\text{att}}(||p_i - \mathbf{x}||^2)$, with $\sigma_{\text{att}}$ a hyperparameter which determines the *locality* of each of the latents. Finally the value matrix is calculated as a learnable linear transformation $\mathbf{W}_v$ of the appearances $\mathbf{A}$, conditioned through FiLM modulation [37] by a second RFF embedding of the relative poses split into scale- and shift modulations: $\mathbf{V}=\mathbf{W}_v\mathbf{A} \odot \mathbf{W}_{v_\alpha}\gamma_{v_\alpha}(\mathbf{A}) + \mathbf{W}_{v_\beta}\gamma_{v_\beta}(\mathbf{A})$. The latents $z_t^\nu$ are optimized for a single state $\nu_t$, whereas the parameters $\theta$ of the ENF backbone - which consist of all the learnable parameters of the linear layers $\mathbf{W}_q, \mathbf{W}_k, \mathbf{W}_v, \mathbf{W}_{v_\alpha}, \mathbf{W}_{v_\beta}$ used to construct $\mathbf{Q}, \mathbf{K}, \mathbf{V}$ - are shared over all states.

The overall architecture consists of a linear layer $\mathbf{W}\mathbb{R}^c \to \mathbb{R}^d$ applied to $\mathbf{c}_i \in \mathbb{R}^c$, followed by a layernorm. After this, the cross attention listed above is applied, followed by three $d$-dim linear layers, the final one mapping to the output dimension $\mathbb{R}^{\text{out}}$.

**Equivariance follows from sharing $\mathbf{Q}, \mathbf{K}, \mathbf{V}$ over equivalence classes**  Note that the latent space of the ENF is equipped with a group action as: $gz_t^\nu = \{(gp_i, \mathbf{a}_i)\}_{i=1}^N$. As an example, SE(2)-equivariance of the ENF follows from bi-invariance of the quantity $\mathbf{a}$ used to construct $\mathbf{Q}$ under the group action:

$$\forall g \in SE(n): \ (p_i, \mathbf{x}) \mapsto (g\,p_i, g\,\mathbf{x}) \ \Leftrightarrow \ p_i^{-1}\mathbf{x} \mapsto (g\,p_i)^{-1}g\,\mathbf{x} = p_i^{-1}g^{-1}g\,\mathbf{x} = p_i^{-1}g^{-1}g. \tag{11}$$

And so, constructing the matrix containing the relative poses of bi-transformed poses and coordinates $(g\mathbf{P})^{-1}g\mathbf{x}$ as $((g\mathbf{P})^{-1}g\mathbf{x})_{i,:} = p_i^{-1}g^{-1}g\mathbf{x} = p_i^{-1}\mathbf{x}$, we trivially have:

$$\forall g \in SE(n): (p_i, \mathbf{x}) \mapsto (g\,p_i, g\,\mathbf{x}) \ \Leftrightarrow \ \mathbf{Q}(\mathbf{A}) \mapsto \mathbf{Q}(g\mathbf{A}) = \mathbf{Q}(\mathbf{A}). \tag{12}$$

# D    Defining additional bi-invariant attributes

Other examples of the bi-invariants attributes that are used in the experiments section are listed here.

*Full rotation symmetries on the 2-sphere* For the global shallow water equations we defined $\mathbf{a}^{\text{SW}}$ as an attribute that is bi-invariant only to rotations over globe's axis, i.e. rotations over $\phi$. In our experiments we also solve diffusion over the sphere, which is fully $\text{SO}(3)$ rotationally symmetric. To achieve equivariance to full 3d rotations, we take poses $p \in \text{SO}(3)$ parameterized by euler angles which act on points $x \in S^2$ parameterized by 3D unit vectors $\mathbf{x}$ through 3D-rotation matrices, allowing us to calculate the bi-invariant $p^{-1}x$:

$$\mathbf{a}^{\text{SO}(3)}_{i,m} = \mathbf{R}_i\mathbf{x}_m. \tag{13}$$

This bi-invariant is used in our experiments for diffusion on the 2-sphere.

*The 3D ball $\mathbb{B}^3$.* We experiment with Boussinesq equation for internally heated convection in a ball. The PDE is fully rotationally symmetric, but since the heat source $K$ is at a fixed point (the center of the ball resp.), it is not symmetric to translations of the initial conditions within the ball. As such, we let $p \in \text{SO}(3) \times \mathbb{R}$ with $\phi, \theta, \gamma, r$ s.t. $0 < r < 1$. The PDE is defined over spherical coordinates $(\phi, \theta, r)$, which we map to vectors in $\mathbf{x} \in \mathbb{R}^3$. We then use the following bi-invariant, which is only symmetric to rotations in $\text{SO}(3)$:

$$\mathbf{a}^{\mathbb{B}^3}_{i,m} = \mathbf{R}_i\mathbf{x}_m \oplus r_{p_i} \oplus r_{x_m}. \tag{14}$$

*No transformation symmetries.* A simple "bi-invariant" for this setting that preserves all geometric information is given by simply concatenating coordinates $p$ with coordinates $x$:

$$\mathbf{a}^{\emptyset}_{i,m} = p_i \oplus x_m \tag{15}$$

Parameterizing the cross-attention operation in Eq. 5 as function of this bi-invariant results in a framework without any equivariance constraints. We use this in experiments to ablate over equivariance constraints and its impact on performance.

# E    Experimental Details

## E.1    Dataset creation

For creating the dataset of PDE solutions we used py-pde [54] for Navier-Stokes and the diffusion equation on the plane. For the shallow-water equation and the diffusion equation on the sphere, as well as the internally heated convection in a 3D ball we used Dedalus [10].

**Diffusion on the plane.**    For the diffusion equation on the plane we use as initial conditions narrow spikes centred at random locations in the left half of the domain for the train set, and in the right half of the domain for the test set. States are defined on a $64 \times 64$ grid ranging from -3 to 3. Initial conditions are randomly sampled uniformly between -2 and 2 for $x$ and 0 and 2 for $y$ in the training set and between -2 and 2 for $x$ and -2 and 0 for $y$. A random value uniformly sampled between 5.0 and 5.5 is inserted at the randomly sampled location. We solve the equation with an Euler solver for 27 steps, discarding the first 7, with a timestep $dt = 0.01$. We generate 1024 training and 128 test trajectories.

**Navier-Stokes on the flat 2-torus.**    For Navier-Stokes on the flat 2-torus we use Gaussian random fields as initial conditions and solve the PDE using a Cranck-Nicholson method with timestep $dt = 1.0$ for 20 steps. The PDE is $\frac{dv}{dt} = -u\nabla v + v\Delta\mu + f, v = \nabla \times u, \nabla u = 0$, where $u$ is the velocity field, $v$ the vorticity, $\mu$ the viscosity and $f$ a forcing term

$$\frac{dv}{dt} = -u\nabla v + v\Delta\mu + f$$
$$v = \nabla \times u$$
$$\nabla u = 0,$$

where $u$ is the velocity field, $v$ the vorticity, $\mu$ the viscosity and $f$ a forcing term. We set viscosity to $1E - 3$, resulting with our setup in a Reynolds number of $\sim \frac{1}{v} = 1000$. States are defined on a $64 \times 64$ grid. We generate 8192 training and 512 test trajectories.

**Diffusion on the 2-sphere.**    For the diffusion dataset on the sphere, states are defined over a $128 \times 64$ $\phi, \theta$ grid. Initial conditions are generated as a gaussian peak inserted at a random point on the sphere with $\sigma = 0.25$. The equation is solved for 20 timesteps with RK4 and $dt = 1.0$. We generate 256 training and 64 test trajectories.

**Spherical whallow-water equations [16].**   The global shallow-water equations are

$$\frac{du}{dt} = -fk \times u - g\nabla h + \nu \Delta u$$

$$\frac{dh}{dt} = -h\nabla \cdot u + \nu \Delta h,$$

where $\frac{d}{dt}$ is the material derivative, $k$ is the unit vector orthogonal to the surface of the sphere, $u$ is the velocity field that is tangent to the spherical surface and and $h$ is the thickness of the fluid layer. The rest are constant parameters of the Earth (see [16] for details). As initial conditions we follow [16] and use basic zonal flow, representing a mid-latitude tropospheric jet, with a correspondingly balanced height field.

$$u(\phi) = \begin{cases} 0 & \text{for } \phi \leq \phi_0 \\ \frac{u_{\max}}{e_n} \exp \left[ \frac{1}{(\phi - \phi_0)(\phi - \phi_1)} \right] & \text{for } \phi_0 < \phi < \phi_1 \\ 0 & \text{for } \phi \geq \phi_1 \end{cases}$$

Where $u_{\max} = 80ms^{-1}$, $\phi_0 = \pi/7$, $\phi_1 = \pi/2 - \phi_1$, and $e_n = \exp[-4(\phi_1 - \phi_0)^2]$. With this initial zonal flow, we numerically integrate the balance equation

$$gh(\phi) = gh_0 - \int^{\phi} au(\phi') \left[ f + \frac{\tan(\phi')}{a} u(\phi') \right] d\phi',$$

to obtain the height $h$. We then randomly generate small un-balanced perturbations $h'$ to the height field

$$h'(\theta, \phi) = \hat{h} \cos(\phi) e^{-(\theta_2 - \theta/\alpha)^2} e^{-[(\phi_2 - \phi)/\beta]^2}$$

by uniformly sampling $\alpha, \beta, \hat{h}, \theta_2$, and $\phi_2$ within a neighbourhood of the values use in [16]. States are defined on a $192 \times 96$ grid for the high-resolution dataset, which is subsequently downsampled by $2 \times 2$ mean pooling to a $96 \times 48$ grid. We generate 512 training trajectories and 64 test trajectories.

**Internally-heated convection in the ball.**   The equations for the internally-heated convection system are listed here, they include thermal diffusivity ($\kappa$) and kinematic viscosity ($\nu$), given by:

$$\kappa = (\text{Ra} \cdot \text{Pr})^{-1/2}$$

$$\nu = \left( \frac{\text{Ra}}{\text{Pr}} \right)^{-1/2}$$

We set Ra $= 1e - 6$ and Pr $= 1$.

1. Incompressibility condition (continuity equation):

$$\nabla \cdot \mathbf{u} + \tau_p = 0$$

2. Momentum equation (Navier-Stokes equation):

$$\frac{\partial \mathbf{u}}{\partial t} - \nu \nabla^2 \mathbf{u} + \nabla p - \mathbf{r}T + \text{lift}(\tau_u) = -\mathbf{u} \times (\nabla \times \mathbf{u})$$

3. Temperature equation:

$$\frac{\partial T}{\partial t} - \kappa \nabla^2 T + \text{lift}(\tau_T) = -\mathbf{u} \cdot \nabla T + \kappa T_{\text{source}}$$

4. Shear stress boundary condition (stress-free condition):

$$\text{Shear Stress} = 0 \text{ on the boundary}$$

5. No penetration boundary condition (radial component of velocity at $r = 1$):

$$\text{radial}(\mathbf{u}(r = 1)) = 0$$

6. Thermal boundary condition (radial gradient of temperature at $r = 1$):

$$\text{radial}(\nabla T(r = 1)) = -2$$

7. Pressure gauge condition:

$$\int p \, dV = 0$$

The boundary conditions imposed are stress-free and no-penetration for the velocity field and a constant thermal flux at the outer boundary. These conditions are enforced using penalty terms ($\tau$) that are lifted into the domain using higher-order basis functions.

States are defined over a $64 \times 24 \times 24$ $\phi, \theta, r$ grid. We use a SBDF2 solver which we constrain by $dt_{\min} = 1e - 4$ and $dt_{\max} = 2e - 2$. We evolve the PDE for 26 timesteps, discarding the first 6. We generate 512 training trajectories and 64 test trajectories.

## E.2 Training details

We provide hyperparameters per experiment. We optimize the weights of the neural field $f_\theta$, and neural ODE $F_\psi$ with Adam [23] with a learning rate of 1E-4 and 1E-3 respectively. We initialize the inner learning rate that we use in Meta-SGD [28] for learning $z^\nu$ at 1.0 for $p$ and 5.0 for $c$. For the neural ODE $F_\psi$, we use 3 of our message passing layers in the architecture specified in [5], with a hidden dimensionality of 128. The std parameter of the RFF embedding functions $\gamma_q, \gamma_{v_\alpha}, \gamma_{v_\beta}$ (see Appx. C), is chosen per experiment. We run all experiments on a single A100. All experiments are ran 3 times.

**Diffusion on the plane.** We use 4 latents with $c \in \mathbb{R}^{16}$. We set the hidden dim of the ENF to 64 and use 2 attention heads. We train the model for 1000 epochs. We set $\gamma_q = 0.05, \gamma_{v_\alpha} = 0.01, \gamma_{v_\beta} = 0.01$. We use a batch size of 8. The model takes approximately 8 hours to train.

**Navier-Stokes on the flat 2-torus.** We use 4 latents with $c \in \mathbb{R}^{16}$. We set the hidden dim of the ENF to 64 and use 2 attention heads. We train the model for 2000 epochs. We set $\gamma_q = 0.05, \gamma_{v_\alpha} = 0.2, \gamma_{v_\beta} = 0.2$. We use a batch size of 4. The model takes approximately 48 hours to train.

**Diffusion on the 2-sphere.** We use 18 latents with $c \in \mathbb{R}^4$. We set the hidden dim of the ENF to 16 and use 2 attention heads. We train the model for 1500 epochs. We set $\gamma_q = 0.01, \gamma_{v_\alpha} = 0.01, \gamma_{v_\beta} = 0.01$. We use a batch size of 2. The model takes approximately 12 hours to train.

**Spherical whallow-water equations [16].** We use 8 latents with $c \in \mathbb{R}^3 2$. We set the hidden dim of the ENF to 128, and use 2 attention heads. We train the model for 1500 epochs. We set $\gamma_q = 0.05, \gamma_{v_\alpha} = 0.2, \gamma_{v_\beta} = 0.2$. We use a batch size of 2. The model takes approximately 24 hours to train.

**Internally-heated convection in the ball** We use 8 latents with $c \in \mathbb{R}^3 2$. We set the hidden dim of the ENF to 128, and use 2 attention heads. We train the model for 1500 epochs. We set $\gamma_q = 0.05, \gamma_{v_\alpha} = 0.2, \gamma_{v_\beta} = 0.2$. We use a batch size of 2. The model takes approximately 24 hours to train.

**CFDBench [32]** We use 25 latents with $c \in \mathbb{R}^{16}$. We set the hidden dim of the ENF to 128, and use 1 attention head. We set $\gamma_q = 0.05, \gamma_{v_\alpha} = 0.1, \gamma_{v_\beta} = 0.1$. We train the model for 100 epochs on a single A100 GPU, taking about 16 hours.

**Baselines** As baseline models on Navier-Stokes we train FNO and GFNO [29] with 8 modes and 32 channels for 700 epochs (until convergence). We train CNODE [2] with 4 layers of size 64 for 300 epochs (until convergence). We train DINo on all experiments for 2000 epochs with an architecture as specified in [51]. For the IHC and shallow-water experiments, we increase the latent dim from 100 to 200, the number of layers for the neural ODE from 3 to 5, and the latent dim of the neural field decoder from 64 to 256, as per [51].

For the Navier-Stokes and Internally-Heated Convection experiments we additionally train the Transolver [50] model. We adapt the Transolver model in PyTorch from the official github repository, and use the same hyperparameter settings as in the original paper (resulting in a 7.1M parameter model) - modifying the training objective to be identical to the autoregressive one we use in our experiments (i.e. mapping from frame to frame). During training in the Navier-Stokes task we observed noticeable instabilities, with the method producing high-frequency artefacts in rollouts in this autoregressive setting. For sparsely subsampled initial conditions, performance deteriorates further - highlighting the benefit of NeF-based continuous PDE solving.

For experiments on internally-heated convection, due to the large size of the input frames, we were required to scale down the Transolver model size in order to be able to fit the model on our A100 GPU, from 256 to 64 hidden units, resulting in a 1.2M parameter model - somewhat comparable in size to the model we use in our experiments (889K). We train the model for 2000 epochs on an A100 GPU, taking approximately 30 hours.

## F   Additional results

**Parameter counts, memory and time efficiency** In order to compare parameter, memory and time efficiency of our method to other baselines, we provide details of our models compared to [51] in Tab. 7, and memory usage/inference time compared to the models we used as baselines in the Navier-Stokes experiments in 8. We note that although our method is not the most memory efficient - meta-learning requires a significant computational overhead - it is the fastest in inference settings when unrolling an unseen state. This is attributable to (1) the fact that we're using a latent-space solver which operates on a drastically compressed representation of the PDE state, and use (2) meta-learning over auto-decoding, which means we only require 3 gradient descent steps to obtain the initial state compared to 300-500 typically used in auto-decoding [51].

Table 7: Parameter count for model used in Navier-Stokes, Shallow-Water and IHC experiments.

| | # PARAMS $f_\theta$ | # PARAMS $F_\psi$ | TOT # PARAMS |
|---|---|---|---|
| | NAVIER-STOKES | | |
| DINo [51] | 333k | 502k | 835k |
| Ours | 354k | 531k | 885k |
| | SHALLOW-WATER | | |
| DINo [51] | 639k | 902k | 1,5M |
| Ours | 356k | 533k | 889k |
| | INTERNALLY-HEATED CONVECTION | | |
| DINo [51] | 639k | 702k | 1,3M |
| Ours | 356k | 533k | 889k |

Table 8: Parameter count and runtimes for for models used in Navier-Stokes experiments. Note that training runtimes are measured per epoch, inference runtimes are measured for unrolling a single 20-step trajectory. GPU memory allocations are measured per trajectory.

| | # PARAMS | RUNTIME/EP | GPU MEM | RUNTIME/TRAJ | GPU MEM |
|---|---|---|---|---|---|
| | | | TRAINING | | INFERENCE |
| FNO | 541k | 40.7s | 0.35Gb | 54.0ms | 0.20Gb |
| GFNO | 3.8M | 200.6s | 2.08Gb | 423.1ms | 1.28Gb |
| CNODE | 1.8M | 1480.0s | 1.45Gb | 170.4ms | 0.71Gb |
| DINo [51] | 835k | 25.8s | 0.23Gb | 61.7ms | 0.14Gb |
| Transolver [50] | 7.1M | 619.2s | 9.42Gb | 43.6ms | 5.19Gb |
| Ours | 885k | 70.0s | 0.70Gb | 9.5ms | 0.61Gb |

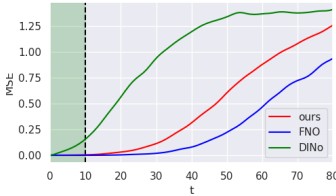

Figure 9: Error accumulation over long rollouts for Navier-Stokes test set with $\nu_0$ 100% observed, train horizon $t_{in}$ is marked in green.

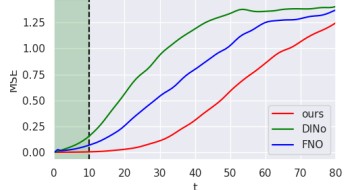

Figure 10: Error accumulation over long rollouts for Navier-Stokes test set with $\nu_0$ 50% observed, train horizon $t_{in}$ is marked in green.

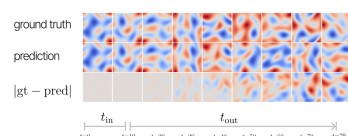

Figure 11: Visualization of an 80-step rollout for a Navier-Stokes test sample.

**Error accumulation on long rollouts for Navier-Stokes** In order to better empirically assess error accumulation of our method for long-term extrapolation, we provide experimental results for unrolling of Navier-Stokes for 80 timesteps. We apply our model both in a setting with fully observed initial conditions and sparsely observed (50%) initial conditions, and provide plots of accumulated MSE along these trajectories in Figs. 9,10, and a visualization of the solution and error in Fig. 11. We compare against DINo [51] and FNO [29], and show that we consistently improve over DINo in terms of long-term extrapolation. FNO achieves better extrapolation limits due to reduced error accumulation in the fully observed setting, but very rapidly deteriorates even within the train horizon with sparsely observed initial conditions, whereas our proposed approach loses very little in terms of extrapolation performance in this sparse setting.

