# OpenReview forum: "Space-Time Continuous PDE Forecasting using Equivariant Neural Fields"
_NeurIPS.cc/2024/Conference — NeurIPS 2024 poster_

### Official Review · Reviewer_zC2e · 2024-07-10

**Soundness:** 3
**Presentation:** 4
**Contribution:** 3
**Rating:** 5
**Confidence:** 4

**Summary:**

The paper presents a novel framework for solving Partial Differential Equations (PDEs) by leveraging the power of Equivariant Neural Fields (ENFs). The authors propose a space-time continuous approach utilizing the symmetry of the PDEs, which is crucial for improving generalization and data-efficiency. The framework is tested on various geometries and PDEs, showing its effectiveness in handling complex dynamics.

**Strengths:**

1. **Data efficiency**: By designing a system that preserves the symmetries of PDEs, the proposed framework enhances the model's ability to generalize from limited data.
2. **Novel initialization method**: The use of meta-learning to structure the latent space of the ENF simplifies the learning process and leads to better performance than autodecoding.

**Weaknesses:**

1. **Error Accumulation:** The usage of ODESolver might pose a challenge with error accumulation over time, particularly for dynamics occurring beyond the training horizon, which could affect the model's long-term predictive accuracy. So it would be helpful if the model is tested in a longer timespan.
2. **Lack of Comparative Analysis:** While the paper compares its approach to a baseline method, a more comprehensive comparison with existing state-of-the-art methods in PDE solving would strengthen the paper's claims, such as Geo-FNO[1], GNOT[2], Transolver[3].

[1] Li, Z., Huang, D. Z., Liu, B., & Anandkumar, A. (2023). Fourier neural operator with learned deformations for pdes on general geometries. *Journal of Machine Learning Research*, *24*(388), 1-26.

[2] Hao, Z., Wang, Z., Su, H., Ying, C., Dong, Y., Liu, S., ... & Zhu, J. (2023, July). Gnot: A general neural operator transformer for operator learning. In *International Conference on Machine Learning* (pp. 12556-12569). PMLR.

[3] Wu, H., Luo, H., Wang, H., Wang, J., & Long, M. (2024). Transolver: A fast transformer solver for pdes on general geometries. *arXiv preprint arXiv:2402.02366*.

**Questions:**

Using ODESolver often incurs higher computational costs. Did the authors implement any acceleration methods to speed up the integration process?

**Limitations:**

1. More experiments should be conducted to prove the model's efficiency in longer timespan.
2. More baselines should be listed in the paper so that model's efficiency can be thoroughly tested and confirmed.

---

> ### Author Rebuttal · Authors · 2024-08-06
>
> We thank the reviewer for their thorough assessment, and for the valuable criticism of our work. We address each of the reviewers' concerns in detail here, and hope to continue the discussion if any details remain unclear.
>
> **Error accumulation** The reviewer is concerned about error accumulation in long-term rollouts as a result of requiring a neural ODE forward propagation. We agree, and as such provide an additional experiment with a more in-depth analysis of long-term rollouts. We extend the dataset of Navier-Stokes solutions we use in the paper to 80 timesteps, and provide test-set results in Figs 1, 2, 3 of the PDF. In Fig. 1 we provide average test MSE per timestep for our model in comparison with FNO and DINo on the setting of having a fully observed initial condition $\nu_0$. Here, we can see that FNO clearly outperforms both NeF-based neural solvers in terms of long-term error accumulation - in agreement with the results shown in our paper in Tab. 1 - although our proposed method retains relatively low MSE up to 20-30 steps after the train horizon (marked in green), where DINo starts error accumulation inside the training time horizon. In Fig. 2 we show results where the initial condition $\nu_0$ for each test trajectory is only sparsely observed, i.e. where only 50% of the initial state is given as input to the model. Here, FNO quickly deteriorates in performance and accumulates error at a higher rate than the NeF-based framework we propose in this work. To us these results indicate that (1) our equivariant NeF-based continuous space-time solving framework provides better generalisation and less error accumulation than its non-equivariant counterpart and (2) retains this relatively limited rate of long-term error accumulation in settings with sparsely observed initial conditions.
>
> **Improving comparative analysis** The reviewer indicates that they would like to see experimental comparison with a wider range of baseline methods. We provide results for a baseline proposed by the reviewer; Transolver, which has shown very promising results for PDE modelling on general geometries.
>
> We took the Transolver model in PyTorch from their github repository, and use the same hyperparameter settings (resulting in a 7.1M parameter model) - modifying the training objective to be identical to the autoregressive one we use in our experiments (i.e. mapping from frame to frame i.o. taking 10 frames as input). Tab. 5 shows the results for the Transolver model on Navier-Stokes in 2D after training on a single A100 to convergence over 700 epochs in approximately 49 hours. Notably, Transolver achieves 1.80E-02 and 1.85E-02 train and test mse respectively within the training horizon, and 4.85E-01 and 4.90E-01 train and test mse outside the training horizon. During training in the Navier-Stokes task we observed noticeable instabilities, with the method producing high-frequency artefacts in rollouts in this autoregressive setting. For sparsely subsampled initial conditions, performance deteriorates further - highlighting the benefit of NeF-based continuous PDE solving.
>
> For experiments on internally-heated convection, due to the large size of the input frames, we were required to scale down the Transolver model size in order to be able to fit the model on our A100 GPU, from 256 to 64 hidden units, resulting in a 1.2M parameter model - somewhat comparable in size to the model we use in our experiments (889K). We train the model for 2000 epochs on an A100 GPU, taking approximately 30 hours. Results in Tab. 4 show the strong performance of this model in the training time horizon - achieving 4.13E-04 test MSE where our framework achieves 5.99E-04, indicating that it is indeed a strong baseline for solving PDEs over complicated geometries. However, we also note that outside of the training horizon, error accumulates more quickly for the Transolver model, indicating it has somewhat overfit the training horizon dynamics. Here, Transolver achieves 2.09E-02 test MSE, where our framework achieves 8.21E-03. We hypothesise that - due to the equivariance constraints placed on our model reflecting inductive biases about this PDE, it extrapolates more reliably beyond the train horizon and over the validation set.
>
> Furthermore, we provide results in Tab. 4 for sparsely observed initial conditions. These results clearly show the advantage of NeF-based continuous solvers in this sparsely observed setting, both DINo and our framework significantly outperform Transolver, which is unable to provide accurate solutions either within the training horizon or outside of it.
>
> In the camera ready version, we will further include the results for the transolver baseline in our other experiments.
>
> **Computational cost of use of ODESolver** The reviewer raises concerns about computational efficiency of our method. We did not implement any specific acceleration methods for solving the forward ODE in the latent space. The main reason we did not feel we need to is because the neural ODE solver operates on a drastically compressed representation of the PDE state, i.e. in the Navier-Stokes experiment we operate on a set of 4 latents with a context vector living in $\mathbb{R}^{16}$. Since the neural ODE solver operates on such a small representation, its overhead is relatively limited. In order to provide more insight into the computational requirements of our method we provide training and inference runtimes and memory usage of our method compared with the different baselines we show in our paper in Tab. 1, 2 of the PDF. We will add these details to the appendix of our manuscript.
>
> We also provide a direction for how to significantly reduce the memory overhead of our decoder by approximating the integral listed in appx C. Eq. 10 in our response to R. hu7A. We will be exploring this in future work.

---

> > ### Comment · Reviewer_zC2e · 2024-08-08
> >
> > It would be better if some results were shown in tabular format. Most of my questions are answered and I will keep my score unchanged.

---

> > > ### Author Response · Authors · 2024-08-08
> > >
> > > We thank the reviewer for their invested time. Tabularized results for the added experiments referred to in our rebuttal text can be found in the PDF attached to the general rebuttal. For the experiments with per-step MSE over 80 unroll steps we felt the error accumulation was better visualized in a graph, however we would be glad to provide those results also in tabular form in the appendix.
> > >
> > > If any questions remain after our rebuttal, we would be happy to discuss them further!

---

### Official Review · Reviewer_hu7A · 2024-07-12

**Soundness:** 3
**Presentation:** 3
**Contribution:** 3
**Rating:** 6
**Confidence:** 3

**Summary:**

This work proposes a space-time continuous method for solving PDEs that respects the inherent symmetries of the PDE via equivariance constraints. Building upon prior work which (a) fits a conditional neural field to output latent vectors and (b) evolves the latent state through time via a Neural ODE, the contribution of this work is to additionally enforce equivariance constraints in the latent space itself. Secondly, the work employs meta-learning to obtain the initial latent representation, which improves the structure of the latent space representation and accelerates the inference time. The authors show improved performance of the method for linear and nonlinear PDEs on complex geometries such as the 2d torus, 3d sphere and 3d ball.

**Strengths:**

- The proposed method significantly reduces overfitting compared to non-equivariant baselines.
 - The method shows good stability at 3–5x the length of the training regime (even though the error accumulates slowly).

**Weaknesses:**

- The computational cost of the method v/s baselines is not shown. Relatedly, do the DINo baselines have similar parameter counts as your method?

**Questions:**

- In figure 2, why is the bottom right image not the rotated version of the solution field on the top right?
 - Complex geometries can be a strong use-case for symmetry-preserving PDEs. However, complex geometries can often have non-trivial boundary conditions as well. Is there any way the method can be extended to handle non-equivariant boundary conditions?
 - Due to the global attention mechanism in the ENF there could be scalability concerns. Could you comment on how the method could be applied to larger scale problems in this case?

---

> ### Author Rebuttal · Authors · 2024-08-06
>
> We thank the reviewer for their assessment of our work, and appreciate their recognition of the benefits of incorporating the symmetry constraints in PDE solving. We elaborate on the raised concerns below.
>
> **Computational cost**. We provide comparison of the computational cost of our method to all other baselines - as well as their parameter counts in tables 1 and 2 of the provided PDF. When looking at the results there can be observed that the training time per epoch is low in comparison to the best performing models (transolver and GFNO), although a bit higher than the less performing models (FNO and DINO). However, we want to stress that looking at inference time, our method is the most time-efficient, mostly attributable to the fact that we’re using a combination of meta-learning with latent-space ODE solving. For new trajectories, initial states only need to be fitted for 3 SGD steps compared to DINO’s 300-500 SGD steps of plane auto-decoding, which provides a clear speed-up during inference.
>
> **Application to non-equivariant boundary conditions** The reviewer asks if/how our method could be extended to non-equivariant boundary conditions. As R. nkRS, i6aX ask the same question, we provide a joint answer in our response to R. nkRS.
>
>  **Scalability and model complexity** The reviewer raises valid concerns regarding the computational cost and scalability of our approach, especially in the context of solving complex/ high-dimensional PDEs. To better assess the computational cost of our approach and contrast it with existing methods, we added information about parameter count, runtimes and GPU memory usage during training and inference (see Tab. 2 of the PDF) for the experiments on Navier-Stokes. These results show that our method excels in inference compared to the baseline methods, although it is not as memory efficient as  FNO[1] or DINo [2]. Although we feel it is outside the scope of our current work, we do agree with the reviewer that aiming for improved scalability should be a focus of future research into this approach, as this would enable application to higher-resolution / higher-dimensional problem settings. We outline an idea for future research into ENF-based PDE solving here, which revolves around a more efficient implementation of the cross-attention operation.
>
> Much of the computational cost of our method is attributable to the calculation of the query and key and values $q_{i,j},k_i, v_{i,j} \in \mathbb{R}^\text{hidden}$ between a point $x_j$ and each of the latents $z_i$ in a latent state $Z$. To sample solution values with a set of $16$ latents for a $64x64$ output grid - for example - this requires the calculation of query and key values for each of $16 \times 64^2$ combinations of latents and points. Note however that, within the ENF architecture’s main operation (described in the manuscript in appx C. Eq 10.) we enforce a latent $ ( p_i, \mathbf{a_{i}} ) $ to be local in the domain of the PDE by weighting the attention value calculated between a latent $i$ and a point $x_j$ through applying a Gaussian window weighted by the relative distance between this point $x_j$ and the latents pose $p_i$ on the domain of the PDE. For latents far removed from a sampled point $x_j$, this Gaussian window forces the attention coefficient from $j$ to $i$ to be negligibly small, in turn nullifying contributions for such latents $i$ to the output for $f_\theta(x_j, Z)$. For an approximation of Eq 10, it could thus be economical to forego calculating these attention coefficients altogether, by first applying an efficient knn algorithm to sort the relative distances from each coordinate $x_j$ to each latent $z_i$, and only calculating query key and value vectors for the nearest $N$ latents. Preliminary results on image regression tasks show that performance loss is negligible, and this method allows for sampling from much larger numbers of latents, making it an interesting possible future direction for scaling the proposed framework to much larger and more complex PDEs.
>
> **In figure 2, why is the bottom right image not the rotated version of the solution field on the top right?** Indeed the function on the bottom right is a rotated version of the solution, but for this particular solution that is quite hard to tell. We will change this figure to be more easily legible in the camera ready version.

---

> > ### Comment · Reviewer_hu7A · 2024-08-13
> > **Response to authors**
> >
> > Thank you for the detailed responses to my questions. I will keep my current score unchanged.

---

### Official Review · Reviewer_i6aX · 2024-07-13

**Soundness:** 2
**Presentation:** 3
**Contribution:** 2
**Rating:** 6
**Confidence:** 3

**Summary:**

The paper attempts to learn the dynamics of certain PDEs from time series data using implicit neural representations, while encoding symmetry information of the domain. In fact, constructing a neural model that is aware of Euclidean transformations is the primary focus of this paper. To this end, the authors design two equivariant neural networks. Given an initial condition, a latent state is obtained by meta-learning. This latent state is then integrated in time by a neural ODE (first network) to obtain a final latent state. The second network then takes this final latent state as an input, and maps any given point coordinate (in the domain) to the solution at the final time. Examples are presented on periodic domains in $ \mathbb{R}^2 $, 2-torus, 2-sphere and the 3D-ball. The paper builds on a 2022 ICLR paper [1] which attempts the same, but without any symmetry assumption.

[1] Yuan Yin, Matthieu Kirchmeyer, Jean-Yves Franceschi, Alain Rakotomamonjy, and Patrick Gallinari. Continuous pde dynamics forecasting with implicit neural representations. 2022.

**Strengths:**

The paper is well written.

PDEs are often posed on domains that have symmetric properties. This is in addition to the fact that the operators appearing in the PDE have their own symmetry / directional properties. While learning from data, most of the existing methods attempting to learn PDE dynamics ignore the symmetry information. Therefore, this is a welcome idea.

The method exhibits impressive extrapolation results.

**Weaknesses:**

Only infinite domain (or periodic boundary conditions) are considered.

In the examples, the transformation groups are chosen by carefully considering the nature of the domain and the operators appearing in the equations. But in a real application, this information, especially the operator information, is not known a priori.

Extrapolation results are shown where results outside the training time horizon are predicted. The problem with such prediction is that they look good until they do not. And there is no logical or analytical bound on the time horizon where the extrapolation is supposed to work. The time horizon is always chosen so as to exhibit the effectiveness of the method. But no analysis is presented in that regard. Therefore such extrapolation results, even though impressive in some respects, do not add to either the understanding or the applicability of this method to a new application.

Memory and execution (training) times are not compared (only the training times of the proposed method are included). Error comparisons are made with other methods. Sometimes this method outperforms the other methods (e.g., Table 2), but in some cases, it is marginally better than the others (e.g., Table 3, 4). Providing memory and training times would make these comparisons more well rounded.

(Minor) Line 276: should be $ \nabla \cdot u = 0 $.

**Questions:**

How will this method work with other boundary conditions?

In the examples, the invariance groups are chosen according to the equation at hand. But how does one choose the invariance groups when the underlying operators and functions (RHS) are unknown?

The extrapolation results are compared with ground truth data, and it is seen that the accuracy deteriorates as the inference horizon goes farther from the  training horizon. In a new application, how to determine a limit for accurate extrapolation, i.e., the temporal horizon where extrapolation is always successful? Does this limit exist?

The forward problem triggers an ODE solve. What is the typical DOFs associated with this ODE solve?

What is the boundary condition applied on the heat equation example?

**Limitations:**

See weaknesses.

---

> ### Author Rebuttal · Authors · 2024-08-06
>
> We thank the reviewer for their very thorough treatment of our manuscript, and appreciate the recognition of the importance of encoding symmetry information in neural PDE solvers. We also acknowledge the reviewer's constructive feedback and address their concerns in detail below.
>
> **How will this method work with other boundary conditions?** In the experiments, we consider both domains like the flat torus (square with periodic boundaries) or the 2-sphere, where no boundary conditions are needed because they are enforced by construction of the bi-invariant, and domains such as the 3-sphere where we use explicit boundary conditions, in particular, zero radial velocity and fixed temperature gradient at the boundary of the sphere. The method proved to work well in both cases.
>
> Other boundary conditions may be incorporated likewise, through appropriate construction of a bi-invariant that respects these boundary conditions. From given boundary conditions it is immediate to verify if they are preserved by a given group action - we show examples of this in the manuscript Sec. 3.1 with Eqs 7, 8. As a further example, for instance for many cylinder flow problems there would exist vertical flip symmetries (the $Z_2$ symmetry group), which could be incorporated by choice of appropriate bi-invariant.
>
> Reviewers R. i6aX, R. NkrS and R. hu7A all ask about the usability in settings when the PDE has no underlying symmetries or when the symmetries are unknown - as such we give an answer jointly. Please see our response to R. NkrS. - under _Equivariance constraints in real-world applications_.
>
> **Limits for successful extrapolation** The reviewer touches upon a very valid drawback of deep-learning based surrogates, namely the fact that to a large extent these methods do not provide any guarantees on extrapolation error bounds as they are purely learned from data with neural networks, and so this would require being able to analytically determine generalisation bounds. We feel this would be a very valuable future research direction.
>
> In order to better empirically assess error accumulation of our method for long-term extrapolation, we provide experimental results for unrolling of Navier-Stokes for 80 i.o. 50 timesteps (see our response to R. zC2e for details). We apply our model both in a setting with fully observed initial conditions and sparsely observed (50%) initial conditions, and provide plots of accumulated MSE along these trajectories in Figs 1, 2, 3. We compare against DINo and FNO, and show that we consistently improve over DINo in terms of long-term extrapolation. FNO achieves better extrapolation limits due to reduced error accumulation in the fully observed setting, but very rapidly deteriorates even within the train horizon with sparsely observed initial conditions, whereas our proposed approach loses very little in terms of extrapolation performance in this sparse setting.
>
> Although we can not provide guarantees on extrapolation limits, results in all experiments show that train and test extrapolation MSEs are relatively consistent. As such, a possible way to obtain empirical extrapolation limits for a new dataset would be to analyse train extrapolation MSE and determine empirical soft bounds for test extrapolation limits by assessing the window in which train extrapolation MSE remains bounded.
>
> **Memory and execution times** We understand that information on the execution / inference times and memory complexity would allow for a better comparison of our approach with baselines. We provide this information in Table 1, 2, which will be included in the appendix of our manuscript! Notably, although our framework has increased GPU memory consumption during training and inference (compared to DINo), we show a marked reduction in inference time attributable to the use of meta-learning to obtain the latents for the initial state.
>
> **What are the typical DOF associated with the ODE solve?** The neural ODE solver operates on the latent space of the ENF, as such, the number of DOF of this ODE is equal to the total size of a single set of latents. We vary this as a hyperparameter over the different datasets:
> - Navier-Stokes: 4 latents, each with a 2-D pose $p_i$ and a 16-D context vector $\mathbf{c}_i$, 72 DOF.
> - Planar diffusion: 4 latents, each with a 2-D pose $p_i$ and a 16-D context vector $\mathbf{c}_i$, 72 DOF.
> - Spherical diffusion: 18 latents, each with a 2-D pose $p_i$ and a 4-D context vector $\mathbf{c}_i$, 108 DOF.
> - Shallow-water: 8 latents, each with a 2-D pose $p_i$ and a 32-D context vector $\mathbf{c}_i$, 272 DOF.
> - Internally heated convection: 25 latents, each with a 3-D pose $p_i$ and a 32-D context vector $\mathbf{c}_i$, 875 DOF.
>
> We typically obtain these settings through running a small hyperparameter search over a range of different settings. We will add these numbers to the appendix for the camera-ready version.
>
> **What is the boundary condition applied in the heat equation example** We use Dirichlet boundary conditions with value 0.

---

> > ### Comment · Reviewer_i6aX · 2024-08-10
> >
> > I thank the authors for answering my questions in detail. I am going to keep my original score unchanged.

---

### Official Review · Reviewer_kxDM · 2024-07-15

**Soundness:** 3
**Presentation:** 3
**Contribution:** 2
**Rating:** 5
**Confidence:** 3

**Summary:**

The work proposes a novel framework combining equivariant neural fields and neural ODEs, providing a continuous space-time solution for PDEs while respecting associated equivariance constraints. The author uses PDE-specific bi-invariant attributes for equivariant neural fields and a meta-learning approach for learning the initial latent state. The proposed method achieves better performance on the chosen PDE problems.

**Strengths:**

1. The work addresses an important and complex issue of equivariance in the context of solving partial differential equations (PDEs). The proposed architecture is not only space-time continuous but also respects the equivariance constraint. This characteristic makes it particularly valuable and effective for various types of scientific research and applications.


2. The proposed method is well-motivated and clearly explained in the paper.

**Weaknesses:**

I have found the empirical study to be the weak point of the work. In order to argue the effectiveness of the proposed solution over existing approaches, the authors need to consider established benchmarks, large-scale datasets, PDEbench, and CFDBench, especially with irregular domains (domains with holes or solid objects as hindrances).

I also find that the choice of baselines is not extensive. For example, SFNO [a] is used for the shallow water equation. Also, baselines like [b,c] are not considered.

Also, as the proposed solution is claimed to be time continuous, zero-shot super-resolution along the time domain should be demonstrated (analogous to Table 3).


a. Spherical Fourier Neural Operators: Learning Stable Dynamics on the Sphere

b. GNOT: A General Neural Operator Transformer for Operator Learning

c. Geometry-Informed Neural Operator for Large-Scale 3D PDEs

**Questions:**

1. What is the training and inference time of the proposed method compared to existing methods like FNOs and Deeponets?

2. what is the Reynolds number of the Navier-Stokes equation problem?

**Limitations:**

yes

---

> ### Author Rebuttal · Authors · 2024-08-06
>
> We thank reviewer kxDM for their effort in reviewing our work. We’re glad to see that the reviewer agrees on the value of adding equivariance constraints to neural PDE solving.
>
> **Improving comparative analysis** The reviewer raised concerns about comparison with a wider set of baselines. To this end, we include a general purpose transformer PDE solving architecture (Transolver [1]) to the list of baseline by applying it both to 2D Navier Stokes and 3D internally heated convection, since this model has shown SOTA results on PDE solving over different geometries.
>
> We adapt the Transolver model in PyTorch from the official github repository, and use the same hyperparameter settings as in the original paper (resulting in a 7.1M parameter model) - modifying the training objective to be identical to the autoregressive one we use in our experiments (i.e. mapping from frame to frame). Tab. 5 shows the results for the Transolver model on Navier-Stokes in 2D. Notably, Transolver achieves 1.80E-02 and 1.85E-02 train and test mse respectively within the training horizon, and 4.85E-01 and 4.90E-01 train and test mse outside the training horizon. During training in the Navier-Stokes task we observed noticeable instabilities, with the method producing high-frequency artefacts in rollouts in this autoregressive setting. For sparsely subsampled initial conditions, performance deteriorates further - highlighting the benefit of NeF-based continuous PDE solving.
>
> For experiments on internally-heated convection, due to the large size of the input frames, we were required to scale down the Transolver model size in order to be able to fit the model on our A100 GPU, from 256 to 64 hidden units, resulting in a 1.2M parameter model - somewhat comparable in size to the model we use in our experiments (889K). We train the model for 2000 epochs on an A100 GPU, taking approximately 30 hours. Results in Tab. 4 show the performance of this model within the training time horizon - achieving 4.13E-04 test MSE where our framework achieves 5.99E-04, indicating that it is indeed a strong baseline for solving PDEs over complicated geometries. However, we also note that outside of the training horizon, error accumulates more quickly for the Transolver model, indicating it has somewhat overfit the training horizon dynamics. Here, Transolver achieves 2.09E-02 test MSE, where our framework achieves 8.21E-03. We hypothesise that - due to the equivariance constraints placed on our model reflecting inductive biases about this PDE, it extrapolates more reliably beyond the train horizon.
>
> We also provide results in Tab. 4 for sparsely observed initial conditions. These results clearly show the advantage of NeF-based continuous solvers in this sparsely observed setting, both DINo and our framework significantly outperform Transolver, which is unable to provide accurate solutions either within the training horizon or outside of it.
>
> **Application to irregular domain** Secondly, the reviewer raised the concern that the method is only tested on regular domains and asked whether we could extend the empirical study to irregular domains as well. We apply our model to the recommended CFDBench dataset [2], which contains solutions to initial conditions for computational fluid dynamics problems with varying boundary conditions, fluid properties and domain shapes - and applied our method on three different problems (cavity, dam and cylinder). We fit these problems with translational bi-invariants (which results in similar translation equivariance constraints to CNN-based neural PDE solvers) - keeping the same model architecture we used in all our experiments, with 25 latents and context vectors $\mathbf{c_i} \in \mathbb{R}^{16}$ and without any specific finetuning. The results can be seen in Tab. 6 in the PDF and show that our approach also handles these more complicated and varied geometries and boundary conditions well - even without the presence of the global symmetries. We hypothesise that the weight-sharing that the equivariance constraints result in might have a regularising effect beneficial to PDE solving; PDEs are built from differential operators that are themselves generally equivariant.
>
> **Continuous-time solving** We provide results for a zero-shot temporal superresolution experiment to show the continuous-time property of our solving framework. We generate a dataset of Navier-Stokes solutions at a higher time-resolution, with timestep size $d\tau=0.25$. We train our model on temporally subsampled frames resulting in a training time resolution of $d\tau=1.0$. We then evaluate the model on step-sizes $d\tau=0.5, d\tau=0.25$. As shown in Tab. 3 of the attached PDF, both inside and outside the train horizon the higher sampling resolution introduces very little error, even with $4\times$ as many unrolling steps, showcasing reliable continuous-time performance of the proposed approach.
>
> **Experimental details / computational requirements** The reviewer asked what the Reynolds number was used to generate the Navier Stokes data. In our experiment we use viscosity $\nu=$ 1e-3, which in our setup results in a Reynolds number of $\frac{1}{\nu} \sim 1000$.
>
> We provide training and inference times for our method compared to the baseline models in Tab. 2 of the PDF. We note that although memory-wise our method has somewhat significant requirements (due to memory-intensive meta-learning), during inference our method outperforms the baselines.
>
> [1] Wu, H., Luo, H., Wang, H., Wang, J., & Long, M. (2024). Transolver: A fast transformer solver for pdes on general geometries. arXiv preprint arXiv:2402.02366.
>
> [2] Luo, Y., Chen, Y., & Zhang, Z. (2023). Cfdbench: A comprehensive benchmark for machine learning methods in fluid dynamics. arXiv preprint arXiv:2310.05963.

---

> > ### Comment · Reviewer_kxDM · 2024-08-12
> >
> > Thanks for the additional experiments. For the experiments on the irregular domain (Table 6, rebuttal pdf), how have you used FNO? FNO is generally only used for regular grids.

---

> > > ### Author Response · Authors · 2024-08-12
> > >
> > > To use FNO in the sparsely observed setting we randomly sample a mask that is used as dropout applied to the initial condition, e.g. 50% of the values of the initial condition are set to zero.
> > >
> > > Thanks for your question! We'll make sure this is clear from the manuscript itself. If any concerns remain, please let us know!

---

### Official Review · Reviewer_NkrS · 2024-07-15

**Soundness:** 3
**Presentation:** 3
**Contribution:** 3
**Rating:** 7
**Confidence:** 4

**Summary:**

The paper introduces a novel framework that leverages Equivariant Neural Fields (ENFs) to solve Partial Differential Equations (PDEs). By preserving geometric information in the latent space, the proposed method respects the known symmetries of the PDE, enhancing generalization and data efficiency. The framework demonstrates improved performance in various challenging geometries, validated through experiments against other neural PDE forecasting methods.

**Strengths:**

The paper presents an innovative approach by integrating equivariant neural fields, which respect the symmetries of PDEs, thereby enhancing model performance.
The methodology addresses significant limitations of existing NeF-based PDE solvers, particularly in generalization to unseen spatial and temporal locations and geometric transformations.
Extensive experimental validation across various geometries (e.g., plane, torus, sphere) demonstrates the robustness of the proposed framework over existing methods.

**Weaknesses:**

The framework's performance decreases when extrapolating beyond the training horizon for complex PDEs.
While the approach shows competitive performance, the computational complexity due to the global attention operator in the ENF backbone can be high.
Error accumulation in long-term predictions could be mitigated with increased model capacity, but this comes at the cost of computational resources.

**Questions:**

None

**Limitations:**

The framework assumes that the boundary conditions are symmetric and that the PDEs exhibit certain symmetries. In real-world applications, these assumptions might not always hold, potentially limiting the applicability of the proposed method to PDEs with different or more complex boundary conditions.

The use of a global attention operator in the Equivariant Neural Field (ENF) backbone increases the computational complexity. This can lead to high computational costs, especially when scaling the model for larger datasets or more complex PDEs.

---

> ### Author Rebuttal · Authors · 2024-08-06
>
> We thank the reviewer for their thorough assessment of our work, and we’re happy to see that the reviewer deems our approach innovative and appreciates the experimental validation we provide. We thank the reviewer for highlighting a number of important considerations with regards to our proposed approach, and discuss limitations mentioned by the reviewer in detail here. We would love to hear your thoughts.
>
> **Equivariance constraints in real-world applications** The reviewer mentions that our method assumes symmetric boundary conditions and/or symmetric PDE formulations, and remarks that in real-world applications, such information might not always be known - limiting applicability of our approach in these cases.
>
> *Note: Since R. i6aX, hu7A also ask after usability with more complex boundary conditions and applicability when PDE symmetries are unknown, we provide a joint response here.*
>
> In general, not knowing the boundary conditions in advance when solving a PDE would result in an ill-posed problem. Boundary conditions help ensure the uniqueness and stability of the solution. From given boundary conditions it is immediate to verify if they are preserved by a given group action - we show examples of this in the manuscript Sec. 3.1 with Eqs 7, 8.
>
> The concern raised about not always knowing the symmetry in advance is a very valid point, but it must be placed within the broader context of inductive biases in deep learning. First, our method extends existing approaches by introducing equivariance - an inductive bias that has shown to be beneficial in many DL-based scientific research applications [3, 4] - to improve forecasting accuracy and consistency for PDEs with known symmetries. Furthermore, our proposed attention architecture with Gaussian windows results in local latent variables that can be beneficial regardless of whether or not symmetries are present. Second, it is often the case - also in real-world applications - that some a priori knowledge about the system is available. It is reasonable to assume, for example, that a spherical domain, such as in the shallow water system, is equivariant with respect to rotations around the axis of rotation. On the contrary, full SO(3) equivariance in real-world climate data defined over the globe can be ruled out a priori by noting that Coriolis (fictitious) forces, due to the rotation of the sphere, would break this symmetry.
>
> Note that our method can indeed be applied when symmetries are unknown or inexact, as we show with an additional experiment applied on CFDBench - as suggested by R. kxDM and shown in Tab. 6. We hypothesise that often the differential operators that define PDEs themselves are symmetric (i.e. derivatives, Laplacians), so introducing these same symmetries into the building blocks of the neural solver itself might still be beneficial to reduce the problem complexity, but an extensive investigation would be needed to properly verify this intuition. Another possibility would be to allow for soft-equivariance constraints to be imposed on the modelled solutions. A simple approach to this end - gleaned from application of GNNs used in latent force field discovery in particle physics [8] - would be to attach one or more symmetry-breaking dataset-wide shared “reference frame nodes” to the latent ODE graph, that could encode for any symmetry-breaking biases that the data contains, to e.g. account for absolute positioning of hindrances or objects within the domain.
>
> Recent works in equivariant deep learning have explored symmetry-discovery / soft symmetry constraints in model design, highlighting another interesting research direction. E.g. [5] initialise their model as being fully equivariant, but allow relaxation of this constraint through optimization, learning an interpolation factor between a set of equivariant and non-equivariant kernels. [6] instead proposes to learn generators for symmetry groups from data directly, overcoming the need for explicit specification of the specific equivariance constraints for novel PDEs - and provides a very extensive list of examples of symmetries common to a variety of PDEs (moreover noting that “simplifying systems of PDEs using their symmetries and consequent coordinate transformations was, in fact, the primary reason Sophus Lie discovered Lie groups”). Though it falls outside of the scope of our current work, similar adaptations might help the proposed framework transfer effectively to settings with inexact or unknown symmetries.
>
>  **Scalability and model complexity** The reviewer remarks that the global attention operation used in the decoder may limit scalability of our method. Since our latents are localised we can approximate the attention operator by restricting the number of latents that are attended to based on their relative distance to the sampled coordinate. We provide a little further detail in our response to R. hu7A.
>
> [1] Li, Z., et al. Fourier Neural Operator for Parametric Partial Differential Equations. In ICLR.
>
> [2] Yin, Y., et al. Continuous PDE Dynamics Forecasting with Implicit Neural Representations. In The Eleventh International Conference on Learning Representations.
>
> [3]  Helwig, J., et al (2023). Group equivariant fourier neural operators for partial differential equations. arXiv preprint arXiv:2306.05697.
>
> [4] Bogatskiy, A., et al. (2022). Symmetry group equivariant architectures for physics. arXiv preprint arXiv:2203.06153.
>
> [5] Wang, R., et al. (2022, June). Approximately equivariant networks for imperfectly symmetric dynamics. In International Conference on Machine Learning (pp. 23078-23091). PMLR.
>
> [6] Gabel, A, et al. (2024). Data-driven Lie point symmetry detection for continuous dynamical systems. Machine Learning: Science and Technology, 5(1), 015037.
>
> [7] Kofinas, M, et al. (2024). Latent field discovery in interacting dynamical systems with neural fields. Advances in Neural Information Processing Systems, 36.

---

> > ### Comment · Reviewer_NkrS · 2024-08-12
> >
> > Thank you for the detailed response. I will keep my original score.

---

### Author Rebuttal · Authors · 2024-08-06

We thank the reviewers for their thorough investigation of our work, and for investing the time to write out valuable criticism. We’re happy to see that reviewers regard our equivariant space-time continuous PDE solving method as valuable and effective for scientific research applications. Additionally, we appreciate the recognition of our efforts to address significant limitations of existing NeF-based solvers, and are happy to see that the reviewers found the work well-motivated and clearly explained.

Though the reviewers do not indicate any critical flaws, they raise - and agree on - a number of concerns, primarily aimed at (1) strengthening the motivation of our method for a wider family of PDEs and boundary conditions (Rev. NkrS, kxDM, hu7A) and (2) expanding the experimental validation of our method by including additional baselines and datasets (Rev. hu7A, zC2e, kxDM, i6aX). We address the most pressing of these concerns in this general response, referring to the attached single-page pdf, and provide additional detailed comments to each reviewer individually.

### **Questions regarding motivation/usability for non-symmetric PDEs / PDEs with unknown symmetries**

R. NkrS  and R. i6aX point out that in our analysis “boundary conditions are symmetric and the chosen PDEs exhibit certain symmetries. In real-world applications, these assumptions might not always hold, potentially limiting the applicability of the proposed method” and that “the invariance groups are chosen according to the equation at hand. But how does one choose the invariance groups when the underlying operators and functions (RHS) are unknown”. This is a valid point, however, we highlight that our method *can* be employed with relaxed or without equivariance constraints both through use of a non-symmetric “bi-invariant” (see appx. D $\mathbf{a}^\emptyset_{i,m}$) and through a procedure we outline in our response to R. NkrS under **Equivariance constraints in real-world applications**, which means that it would retain its utility in scenarios where the symmetries are not (fully) known or are deemed irrelevant.

We argue that the attention-based conditional neural field trained with meta-learning can enhance forecasting accuracy due to its ability to capture local dependencies effectively, also in these settings. To underscore this, we provide additional experimental results on CFDBench (see response to R. kxDM), which doesn’t exhibit symmetric boundary conditions or global symmetries and contains varying geometries with hindrances and varying boundary conditions - achieving performance competitive to the baselines set by the authors.

On the other hand, in science and engineering it is not rare that known symmetries are used to develop models even when the complete details of the system are not fully known (e.g., in atmospheric and oceanic sciences rotational symmetry is often assumed). For these reasons, we posit that an equivariant model for spatio-temporally continuous PDE dynamics forecasting constitutes a valuable contribution, also in real-world settings.

### **Improving comparative analysis (R. hu7A, zC2e, kxDM, i6aX)**

**Comparison with additional baselines** Rev. zC2e and kxDM point out that on a number of experiments we compare with Yin. (2023), and argue for including additional baseline results for neural PDE solvers that can similarly handle the varying geometries we’re operating on, as this would strengthen the paper’s claims. To this end, we implemented an additional baseline proposed by the reviewers, which we detail below.

We adopt the Transolver model (2024), and train it on the Navier-Stokes and Internally-Heated Convection tasks. Find a detailed description of the experimental setup in our response to Rev. kxDM, results are listed in Tab 4, 5. The Transolver proved unstable in the autoregressive Navier-Stokes experiment, obtaining comparable performance to DINo, due to high-frequency artefacts arising on rollout - and so it underperforms FNO and our method. Moreover, it deteriorates heavily with sparsely observed initial conditions. Transolver does achieve competitive error in the internally-heated convection experiment in the fully observed setting, i.e. when 100% of the initial state is observed, indicating that it is a strong baseline in settings with complicated geometry and boundary conditions. However, the Transolver model deteriorates considerably in performance with subsampled initial conditions in both experiments - indicating that indeed it is not a model suitable for space-time continuous solving on complex geometries. In these settings, the proposed model architecture outperforms Transolver. We feel these results further strengthen the case for our proposed model architecture, and we thank the reviewers for this suggestion. In the camera ready version, we will further include results for this baseline in our other experiments.

**Computational cost / parameter count** (R. i6aX, hu7A) We provide the parameter counts of our model versus each of the baselines, as well as their respective runtime and GPU memory usage during training and inference in tables 1 and 2 of the pdf. We note that although our method is not the most memory efficient - meta-learning requires a significant computational overhead - it is the fastest in inference settings when unrolling an unseen state. This is attributable to (1) the fact that we’re using a latent-space solver which operates on a drastically compressed representation of the PDE state, and use (2) meta-learning over auto-decoding, which means we only require 3 gradient descent steps to obtain the initial state compared to 300-500 typically used in auto-decoding (Yin 2023).

This information, as well as all extra provided experiments, will be included in the appendix of the manuscript. We again thank the reviewers for their diligence, and hope to discuss further if any questions or concerns remain.

---

### Author Response · Authors · 2024-08-12

We would like to thank the reviewers for their part in improving our manuscript through their dilligence.

As the discussion phase nears its end, we would like to ask you (the reviewers) that have not done so to let us know if our responses addressed your concerns and - if so - whether you would reconsider your initial recommendation.

If you still have concerns, please share so we may discuss and address them.

---

### Decision · Program_Chairs · 2024-09-25

**Decision:**

Accept (poster)

**Comment:**

The paper introduces an encode-process-decode architecture for modeling physical spatio-temporal dynamics using implicit neural representations (INRs), also known as neural fields, that respect the symmetries of physical problems. The primary focus and novelty of the paper lie in the preservation of these symmetry properties within the problem's geometry. This is achieved by incorporating equivariance constraints at the level of the latent representation via equivariant neural fields and within the "process" component. Specifically, the neural ODE solver, implemented using a graph-based message-passing architecture, is constrained to be equivariant to symmetries. The experiments demonstrate the advantages of these constraints across diverse benchmarks.

Building on previous INR solvers, the paper introduces symmetry constraints both in the signal representation and in the solver. Its main contribution is the introduction of equivariance properties within this framework. In particular, the authors adapt the design of equivariant neural fields to the context of partial differential equations (PDEs). The results highlight the importance of such prior physics-informed characteristics for generalizing in time and space for physical dynamical processes. Following the reviewers' comments, the authors included additional results with baselines and datasets during the rebuttal.

Although the paper presents a somewhat incremental contribution, the reviewers regard it as a solid and significant advancement that improves the state of the art. It demonstrates the importance of imposing physics constraints to help neural physics models generalize, particularly from limited data and across diverse geometries. The adaptation of equivariant INR to this domain is novel and contributes to the development of the emerging trend of INR-based PDE solvers.